



# Coral calcifying fluid aragonite saturation states derived from Raman spectroscopy

Thomas M. DeCarlo[1,2], Juan P. D'Olivo[1,2], Taryn Foster[3], Michael Holcomb[1,2], Thomas Becker[4,5], and Malcolm T. McCulloch[1,2]

[1]Oceans Institute and School of Earth Sciences, The University of Western Australia, 35 Stirling Hwy, Crawley 6009, Australia
[2]ARC Centre of Excellence for Coral Reef Studies, The University of Western Australia, 35 Stirling Hwy, Crawley 6009, Australia
[3]Australian Institute of Marine Science, Crawley 6009, Australia
[4]Centre for Microscopy, Characterisation and Analysis, The University of Western Australia, Crawley 6009, Australia
[5]Department of Chemistry, Curtin Institute of Functional Molecules and Interfaces, Curtin University, GPO Box U1987, Perth 6845, Australia

*Correspondence to:* Thomas M. DeCarlo (thomas.decarlo@uwa.edu.au)

**Abstract.** Quantifying the saturation state of aragonite ($\Omega_{Ar}$) within the calcifying fluid of corals is critical for understanding their biomineralisation process and sensitivity to environmental changes including ocean acidification. Recent advances in microscopy, microprobes, and isotope geochemistry allow determination of calcifying fluid pH and $[CO_3^{2-}]$, but direct quantification of $\Omega_{Ar}$ (where $\Omega_{Ar}=[CO_3^{2-}][Ca^{2+}]/K_{sp}$) has proved elusive. Here we test a new technique for deriving $\Omega_{Ar}$ based on

Raman spectroscopy. First, we analysed abiogenic aragonite crystals precipitated under a range of $\Omega_{Ar}$ from 10 to 34, and found a strong dependence of Raman peak width on $\Omega_{Ar}$ that was independent of other factors including pH, Mg/Ca partitioning, and temperature. Validation of our Raman technique for corals is difficult because there are presently no direct measurements of calcifying fluid $\Omega_{Ar}$ available for comparison. However, Raman analysis of the international coral standard JCp-1 produced $\Omega_{Ar}$ of $12.3 \pm 0.3$, which we demonstrate is consistent with published skeletal Sr/Ca, Mg/Ca, B/Ca, $\delta^{44}Ca$, and $\delta^{11}B$ data.

Raman measurements are rapid ($\leq 1$ s), high-resolution ($< 1$ $\mu$m), precise (derived $\Omega_{Ar} \pm 1$ to 2), and require minimal sample preparation; making the technique well suited for testing the sensitivity of coral calcifying fluid $\Omega_{Ar}$ to ocean acidification and warming using samples from natural and laboratory settings. To demonstrate this, we also show a high-resolution time series of $\Omega_{Ar}$ over multiple years of growth in a *Porites* skeleton from the Great Barrier Reef, and we evaluate the response of $\Omega_{Ar}$ in juvenile *Acropora* cultured under elevated $CO_2$ and temperature.

## 1  Introduction

The calcium carbonate ($CaCO_3$) skeletons built by coral polyps are the building blocks of massive coral reef structures that protect shorelines, bolster tourism, and host some of the greatest concentrations of biodiversity on the planet (Knowlton et al., 2010; Costanza et al., 2014). Critical to the coral calcification process is the extraction of $Ca^{2+}$ and $CO_3^{2-}$ ions from seawater to grow aragonitic $CaCO_3$ crystals. But corals today live in seawater that is less conducive to $CaCO_3$ nucleation than it was



just a century ago. In surface waters of the tropical oceans, carbonate ion concentrations ($[CO_3^{2-}]$) have declined by ~15% since 1900 due to invasion of anthropogenic $CO_2$, which dissociates into carbonic acid and decreases seawater pH and $[CO_3^{2-}]$ through a process referred to as ocean acidification (Caldeira and Wickett, 2003; Doney et al., 2009). If anthropogenic $CO_2$ emissions continue unabated, by the end of the 21st century the $[CO_3^{2-}]$ of surface seawater is projected to decline to ~50% of

pre-industrial levels (Hoegh-Guldberg et al., 2014). This rapid change in ocean carbonate chemistry, likely unprecedented for hundreds of millions of years (Hönisch et al., 2012; Zeebe et al., 2016), has sparked concerns for coral growth. Indeed, laboratory experiments repeatedly demonstrate that coral calcification rates decrease in response to lower $[CO_3^{2-}]$ (Gattuso et al., 1998; Chan and Connolly, 2013; Comeau et al., 2017), leading to projections that as $CO_2$ levels continue to rise calcification will decline to unsustainable levels, such that there is net reef erosion (Hoegh-Guldberg et al., 2007; Pandolfi et al., 2011).

Yet signs of resilience do exist. Some coral species are able to maintain normal calcification rates across large natural acidification gradients (Fabricius et al., 2011; Shamberger et al., 2014; Barkley et al., 2015), indicative of adaptation or acclimation (Barkley et al., 2017). One potential mechanism to counteract acidification of surrounding seawater is pH homeostasis at the site of calcification (Georgiou et al., 2015; Barkley et al., 2017). Corals calcify from an isolated fluid located between the living tissue and the existing skeleton (Barnes, 1970; Cohen and McConnaughey, 2003; Venn et al., 2011; Tambutté et al., 2012).

Up-regulation of pH within this fluid, potentially achieved via proton pumping and/or symbiont photosynthesis, elevates the saturation state with respect to aragonite ($\Omega_{Ar}$), driving rapid nucleation and growth of aragonite crystals (Gattuso et al., 1999; Al-Horani et al., 2003; McCulloch et al., 2012). Ultimately, it is the $\Omega_{Ar}$ within the calcifying fluid that determines the rates of crystal growth (Burton and Walter, 1987; Cohen and Holcomb, 2009), and the sensitivity of corals to ocean acidification likely depends on the relationship (if one exists) between external and internal $\Omega_{Ar}$ (McCulloch et al., 2012; Kubota et al., 2015;

Comeau et al., 2017).

      Characterising the $\Omega_{Ar}$ of the calcifying fluid, and understanding its sensitivities to variations in the reef environment, is therefore essential for accurately forecasting coral calcification responses to 21st century ocean acidification. Observing this fluid has proved difficult though, due to its small size and isolation beneath the living polyp (Clode and Marshall, 2002). Estimates of fluid carbonate chemistry have so far been derived from micro-electrodes, pH-sensitive dyes, boron isotopes,

B/Ca, U/Ca, and bulk calcification rates (Al-Horani et al., 2003; Trotter et al., 2011; Venn et al., 2011; DeCarlo et al., 2015; Holcomb et al., 2016; Raybaud et al., 2017). However, these approaches do not always agree (Ries, 2011; Holcomb et al., 2014), and they have so far focused on calcifying fluid carbonate chemistry without considering the effect of $[Ca^{2+}]$ on $\Omega_{Ar}$.

      A potential alternative approach to quantify calcifying fluid $\Omega_{Ar}$ is based on the Raman scattering from a laser focused onto the skeleton. When light interacts with a material, a small percent (typically <0.0001%) of the photons are scattered

inelastically (referred to as Raman or Stokes scattering), resulting in a change of energy and frequency (Smith and Dent, 2005). The frequency shifts associated with Raman scattering are characteristic of both the internal vibrations of a molecule and the lattice vibrations between molecules in a crystal, which makes Raman spectroscopy a valuable tool for mineral identification (Urmos et al., 1991; Dandeu et al., 2006; Brahmi et al., 2010; Clode et al., 2011; Nehrke et al., 2011; Stock et al., 2012; Foster and Clode, 2016; Stolarski et al., 2016; Roger et al., 2017). Importantly, Raman peaks can also provide information regarding

the chemical composition of crystals and the conditions of the fluid from which they formed. For example, in abiogenic




CaCO$_3$, the shapes and positions of the $\nu_1$ peak at ~1085 cm$^{-1}$ (which represents symmetric stretching of the carbonate C-O bond) have been correlated with Mg content and/or crystallinity (Bischoff et al., 1985; Wang et al., 2012; Perrin et al., 2016). A highly crystalline CaCO$_3$ with relatively few defects or impurities will have a narrow $\nu_1$ peak because the C-O bonds throughout the crystals are of the same, or very similar, strength (Bischoff et al., 1985). Defects in the crystals and/or trace element impurities cause positional disorder of CO$_3$ in the lattice. Positional disorder affects the length, and thus the strength and vibrational frequency, of C-O bonds (Bischoff et al., 1985). Stronger bonds and lighter atoms are associated with higher frequency molecular vibrations, while weaker bonds and heavier atoms are associated with lower frequency vibrations (Smith and Dent, 2005; De La Pierre et al., 2014), and these differences shift the wavenumber of the $\nu_1$ peak accordingly (Bischoff et al., 1985). An increase in disorder leads to an increase in the distribution of C-O bond strengths, causing an increase in $\nu_1$ peak width (Bischoff et al., 1985; Addadi et al., 2003; Lin et al., 2007; Wang et al., 2012; McElderry et al., 2013; Perrin et al., 2016).

In biogenic calcium carbonates, correlations have been reported between $\nu_1$ peak widths and environmental conditions, including temperature and seawater pCO$_2$ (Kamenos et al., 2013, 2016; Hennige et al., 2015; Pauly et al., 2015). These changes potentially reflect differences in calcifying fluid carbonate chemistry. Crystals growing from more supersaturated solutions generally have more defects and incorporate more impurities (Watson, 2004), resulting in relatively disordered crystal lattices and wide Raman peaks (Urmos et al., 1991). This hypothesis is generally consistent with empirical observations of biogenic calcium carbonates, in which $\nu_1$ peak widths increase as ambient seawater pCO$_2$ decreases and/or pH increases (*i.e.* higher $\Omega_{Ar}$) (Kamenos et al., 2013; Hennige et al., 2015; Pauly et al., 2015). However, corals exert strong control on the carbonate chemistry of their calcifying fluid by elevating pH and/or $\Omega_{Ar}$ at the site of calcification to facilitate more rapid crystal growth (Al-Horani et al., 2003; Venn et al., 2011; McCulloch et al., 2012). Thus, the observed correlations between Raman peak widths and ambient seawater conditions likely do not reflect the true sensitivity of aragonite Raman peaks to seawater chemistry, but rather the sensitivity of calcifying fluid chemistry to external pH and/or $\Omega_{Ar}$. For this reason, information regarding the relationship between carbonate chemistry and Raman peak width in abiogenic experiments is required to quantitatively interpret Raman spectra of biogenic CaCO$_3$ with respect to the actual calcifying fluid conditions.

Here, we first evaluate the controls on Raman $\nu_1$ peak width using abiogenically precipitated aragonites. These samples allow us to test the sensitivities of $\nu_1$ peak width against geochemical composition (*e.g.* Mg/Ca) and fluid conditions (including $\Omega_{Ar}$, pH and temperature) directly, without the confounding influence of a coral polyp. We derive a calibration between $\nu_1$ peak width and $\Omega_{Ar}$, and apply it to estimate $\Omega_{Ar}$ of the international coral skeleton standard JCp-1, for which independent lines of geochemical evidence allow us to test the accuracy of our approach. Finally, we demonstrate the applicability of Raman spectroscopy by (1) reconstructing multiple years of $\Omega_{Ar}$ variability in a *Porites* coral collected from the Great Barrier Reef (GBR) (D'Olivo and McCulloch, 2017), and (2) comparing $\Omega_{Ar}$ in cultured juvenile *Acropora spicifera* exposed to elevated CO$_2$ and temperature treatments (Foster et al., 2015).





## 2 Methods

### 2.1 Raman Measurements

Raman spectroscopy was used to analyse the abiogenic aragonite precipitates described in DeCarlo et al. (2015) and Holcomb
et al. (2016) (Supplement Table S1). Briefly, aragonite was precipitated by addition of $Na_2CO_3$ and $NaHCO_3$ solutions to
seawater. Various $Na_2CO_3/NaHCO_3$ ratios and pumping rates produced a range of $\Omega_{Ar}$ from 10-34 while achieving some
independence between $\Omega_{Ar}$ and other carbonate system variables (*e.g.* $r^2$ between $\Omega_{Ar}$ and pH was only 0.34). Most (22 of
28) experiments were conducted at 25.5 °C, but two experiments were conducted each at 20 °C, 33 °C, and 40 °C.

Raman spectra of the abiogenic aragonites were originally reported in DeCarlo et al. (2015) and Holcomb et al. (2016).
These initial spectra were collected at Woods Hole Oceanographic Institution (WHOI) with a Horiba LabRam HR800 Raman
spectrometer using a 785 nm laser source, 40x objective, 600 $mm^{-1}$ grating, and CCD detector maintained at -70 °C. Spectral
resolution was approximately 1.2 $cm^{-1}$. Three grains were analysed per experiment with an integration time of 5 s per spec-
trum. Subsequently, we repeated analyses of a subset of these aragonites (samples f02, f03, f06, g07, g13, h09) at the Centre
for Microscopy, Characterisation, and Analysis (CMCA) at the University of Western Australia. These more refined measure-
ments, which are the basis of this study, were performed with a WITec Alpha300 RA+ confocal Raman microscope using a
785 nm laser source, 20x objective with numerical aperture of 0.5, 1200 $mm^{-1}$ grating, and an Andor iDUS 401 CCD detector
maintained at -60 °C. The nominal spectral resolution was 1.3 $cm^{-1}$. A silicon chip was analysed to facilitate comparison of
peak position with other laboratories (the strong Si peak was present at 522.9 $cm^{-1}$). For analysis of the abiogenic aragonites,
at least five grains were analysed per experiment with 5-10 spectra per grain and integration times between 0.1 and 5 s (see
Supplement for a comparison between instruments and discussion of the general applicability of our results to measurements
in other laboratories). All analyses of coral skeletons described below were conducted with the WITec instrument using 1 s
integration times.

The *Porites* coral skeleton standard JCp-1 (Okai et al., 2002) was also analysed to facilitate comparison between Raman
results and geochemical estimates of calcifying fluid $\Omega_{Ar}$ and $[Ca^{2+}]$. We collected 440 spectra from various grains spread
onto a glass slide. Although JCp-1 exists in ground form, we note that analysis of intact coral skeletons is also possible (Wall
and Nehrke, 2012; Hennige et al., 2014) and that peak widths apparently do not depend on whether the sample is ground or
intact (Zakaria et al., 2008), as long as the grain size exceeds the laser spot size.

We also collected Raman spectra down-core in a *Porites* skeleton collected from the Great Barrier Reef (the sample is de-
scribed in detail in D'Olivo and McCulloch (2017). Briefly, the core was collected near Havannah Island, an in-shore reef
in the central GBR where corals bleached during 1998. The skeleton was cut in a slab and cleaned with bleach as described
by D'Olivo and McCulloch (2017), but no additional preparation was required. D'Olivo and McCulloch (2017) present $\delta^{11}B$
and trace element/Ca ratios from multiple tracks in the skeleton, and they show anomalies to the skeletal geochemistry corre-
sponding to the 1998 thermal stress event. We used the automated microscope stage and area-mapping features of the WITec
instrument to measure Raman spectra in a down-core transect at 50-$\mu$m resolution, covering approximately 6 years of growth
from 1996 to 2002. The TrueSurface module was used to construct a topography map of the skeleton that was then followed





by the instrument to ensure the sample was always in-focus for the Raman measurements. A maximum depth of 1 mm below the cut surface was set to limit the integration of signal from skeleton of different ages. Each time-point represents the average of 10 spectra spaced horizontally in 100 $\mu$m increments.

Finally, we analysed skeletons of the cultured juvenile *Acropora* described in Foster et al. (2015, 2016). Briefly, larval planulae were maintained in "Control" (24 °C, 250 $\mu$atm $CO_2$), "+$T$" (27 °C, 250 $\mu$atm $CO_2$), "+$CO_2$" (24 °C, 900 $\mu$atm $CO_2$), and "+$CO_2$+$T$" (27 °C, 900 $\mu$atm $CO_2$) treatments. Raman data were previously reported for these corals in Foster and Clode (2016) but those earlier measurements were conducted with a 600 mm$^{-1}$ grating, which made it difficult to resolve changes in peak width (see Supplement). We therefore collected 25 new Raman spectra from each of three broken skeletal pieces from each of three corals, per treatment. The TrueSurface module was used to ensure all measurements were taken with the optics well focused on the samples.

## 2.2 Calculations and Statistics

For all Raman spectra, we utilised the Full Width at Half Maximum intensity (FWHM) of the $\nu_1$ peak (Fig. 1). A Gaussian-shaped curve was fitted to each spectrum between 1080 and 1100 cm$^{-1}$ with the following expression:

$$y = b + mx + ke^{\frac{-(p-x)^2}{2s^2}} \tag{1}$$

where $y$ is the fit, $x$ is the Raman shift between 1080 and 1100 cm$^{-1}$, $b$ is the background intensity, $m$ is the background slope, $k$ is the peak height, $p$ is the peak position (*i.e.* wavenumber), and $s$ is the standard deviation. Peak intensity varies depending on the smoothness of the surface and the focus of the laser. We removed spectra with $\nu_1$ peak height < 30 intensity units (signal:noise of approximately 5) due to the larger uncertainties associated with curves fit to these spectra. FWHM is calculated following Weisstein (2017):

$$\text{FWHM} = 2s\sqrt{2\ln(2)} \approx 2.3548s \tag{2}$$

where $s$ is the standard deviation of the Gaussian curve from equation (1). Measured Raman peak widths are known to be convolutions of the "true" peak widths and instrument noise (Nasdala et al., 2001; Wang et al., 2012; Váczi, 2014). The effect of instrument noise on Raman peak widths is directly related to the spectral resolution, which must be accounted for when comparing measurements conducted on different instruments (see Supplement). We used the Nasdala et al. (2001) formula to account for the effect of spectral resolution and calculate true FWHM:

$$b_t = b_m\sqrt{1 - 2(\frac{r_s}{b_m})^2} \tag{3}$$

where $b_t$ is the true peak width, $b_m$ is the measured peak width, and $r_s$ is the spectral resolution.



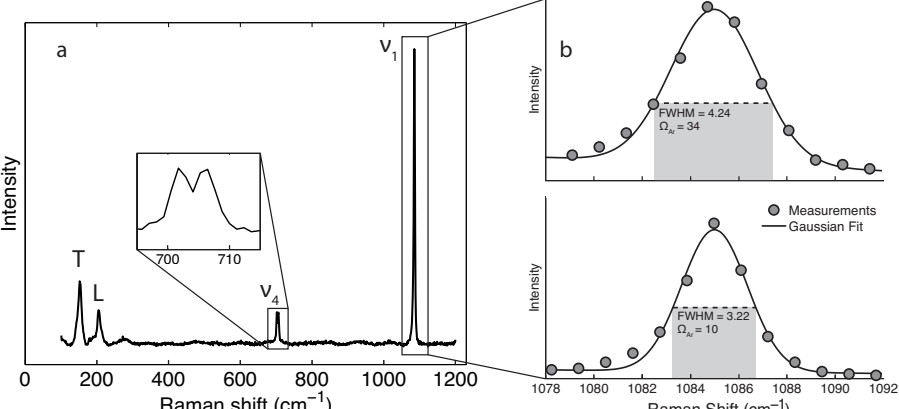

**Figure 1.** Example of Raman spectrum and peak width analysis. (a) Raman spectrum of precipitate from experiment "f03", with the peaks labelled following White (1974) and Bischoff et al. (1985): T = translation (lattice mode), L = libration (lattice mode), $\nu_4$ = in-plane bend (internal mode), and $\nu_1$ = symmetric stretch (internal mode). This spectrum is readily identifiable as $CaCO_3$ based on the strong $\nu_1$ peak at approximately 1085 cm$^{-1}$, and distinguished as aragonite rather than other $CaCO_3$ polymorphs (calcite and vaterite) based on the positions of the T, L, and $\nu_4$ peaks (Dandeu et al., 2006). The double peak between 700-710 cm$^{-1}$ is the feature that most clearly distinguishes aragonite from calcite or vaterite (Urmos et al., 1991). (b) Measurement of $\nu_1$ peak width. The grey points are the measured intensities at the corresponding Raman wavenumber shift. The black lines show the Gaussian curves fit to the data, and the grey boxes indicate the measured peak widths ($\nu_1$ FWHM). Note that measured $\nu_1$ FWHM is converted to true $\nu_1$ FWHM for analysis in subsequent figures. The y-axis scale is in arbitrary intensity units.

In the analysis of the abiogenic aragonite, we used the linear model function in R (R Core Team, 2016) to test for relationships between $\nu_1$ FWHM and $\Omega_{Ar}$, aragonite precipitation rate, aragonite Mg/Ca, fluid $[CO_3^{2-}]$, fluid pH, Raman peak height ($k$), and Raman peak position ($p$). Since $\Omega_{Ar}$ was the single variable most strongly correlated with $\nu_1$ FWHM, we also evaluated linear models with two explanatory variables: $\Omega_{Ar}$ and the rest of the variables listed above. Analysis of variance (ANOVA) was used to test whether multivariate models were significantly better than the model based on $\Omega_{Ar}$ alone. We tested for normality of residuals with Kolmogorov-Smirnov tests and homogeneity of variances with Levene's test.

For the cultured *Acropora*, we calculated the mean $\Omega_{Ar}$ of each skeletal fragment, weighting the 25 measurements per fragment by peak height. We then tested whether the data were significantly different from normal distributions with Kolmogorov-Smirnov tests and homogeneity of variances was checked with Levene's test. Finding no significant differences from normal distributions and no significant differences in variance among treatments, we conducted a two-way ANOVA and evaluated the effects with Tukey's honest significant difference test. Statistical significance was defined as $p < 0.05$.

## 2.3 Mg/Ca partitioning between aragonite and seawater

We also report here the partitioning of Mg/Ca between these abiogenic aragonites and seawater because (1) previous studies have suggested that Mg/Ca is important for interpreting Raman peak widths (Bischoff et al., 1985; Urmos et al., 1991), and





(2) Mg/Ca is used as a constraint in our comparison of JCp-1 geochemistry and Raman spectroscopy. Partitioning of $Mg^{2+}$ between aragonite and seawater can be described by an exchange for $Ca^{2+}$ in the aragonite lattice, with a partition coefficient $(K_D)$ expressed as:

$$K_D^{\text{Mg/Ca}} = \frac{\frac{\text{Mg}}{\text{Ca}}\,_{aragonite}}{\frac{\text{Mg}}{\text{Ca}}\,_{seawater}} \qquad (4)$$

where $K_D$ is dimensionless (Kinsman and Holland, 1969; Gaetani and Cohen, 2006). While there is some uncertainty in the mechanism of $Mg^{2+}$ incorporation into aragonite (Montagna et al., 2014), the $K_D$ nevertheless serves as an empirical measure of the distribution of Mg/Ca between aragonite and seawater under various conditions. DeCarlo et al. (2015) reported $K_D$ for Sr/Ca from these same samples, and here we follow the same calculations for $K_D^{\text{Mg/Ca}}$ using the Mg/Ca data from Holcomb et al. (2016), except for the slight modifications to the calculations described below. Briefly, $K_D^{\text{Mg/Ca}}$ was calculated

from the measured Mg/Ca in the bulk precipitate (Holcomb et al., 2016), estimates of Mg/Ca in the initial fluid (seawater), and modelling the evolution of elemental concentrations in the fluid through the course of each experiment. However, since [Mg] in the initial seawater was not reported, we instead calculated initial [Mg] and [Ca] from the established relationships between salinity and concentrations of these elements in seawater (Riley and Tongudai, 1967). Our approach also differs from DeCarlo et al. (2015) in that they combined element/Ca measurements of both the final solution and the bulk solid to calculate $K_D$,

whereas we used only the reported aragonite Mg/Ca because final fluid [Mg] was not measured. Two of the experiments (f08 and g13) were conducted with initial elemental concentrations modified from seawater by addition of dissolved $CaCO_3$ and $SrCO_3$, and they were excluded from $K_D^{\text{Mg/Ca}}$ calculations due to uncertainty of the fluid Mg/Ca ratio.

## 3 Results

In the abiogenic aragonites analysed in this study, $\nu_1$ FWHM was strongly correlated with seawater $\Omega_{Ar}$ ($r^2$=0.70, p<0.001;

Fig. 2 and Tables 1-2). Aragonite precipitation rate $(G)$ is a function of $\Omega_{Ar}$ and temperature $(T)$ (Burton and Walter, 1987), and thus the experiments conducted at different $T$ allow us to isolate the influence of $\Omega_{Ar}$ from $G$. In contrast to the strong dependence of $\nu_1$ FWHM on $\Omega_{Ar}$, there was no significant correlation between $\nu_1$ FWHM and $G$ (Fig. 3; Table 1). $\nu_1$ FWHM was significantly correlated with the aragonite Mg/Ca ratio, fluid $[CO_3^{2-}]$, pH, $T$, and $\nu_1$ peak height (Fig. 3; Table 1). However, these correlations were all weaker than the correlation between $\nu_1$ FWHM and $\Omega_{Ar}$, and in multivariate models combining

$\Omega_{Ar}$ with either Mg/Ca, $[CO_3^{2-}]$, pH, $T$, or $\nu_1$ peak height, the only significant variable was $\Omega_{Ar}$ (Table 1). Together, this indicates that $\Omega_{Ar}$ is likely the variable controlling $\nu_1$ FWHM, and that the correlations between $\nu_1$ FWHM and other factors arise as artefacts of the correlations between those variables and $\Omega_{Ar}$. The relationship between $\Omega_{Ar}$ and $\nu_1$ FWHM is shown in Table 2 for measured and true FWHM on both the Horiba and WITec instruments. The distributions of residuals were not significantly different from normal distributions (Kolmogorov-Smirnov test, p > 0.8). FWHM data for each experiment are

listed in Supplement Table S2.

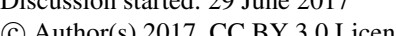



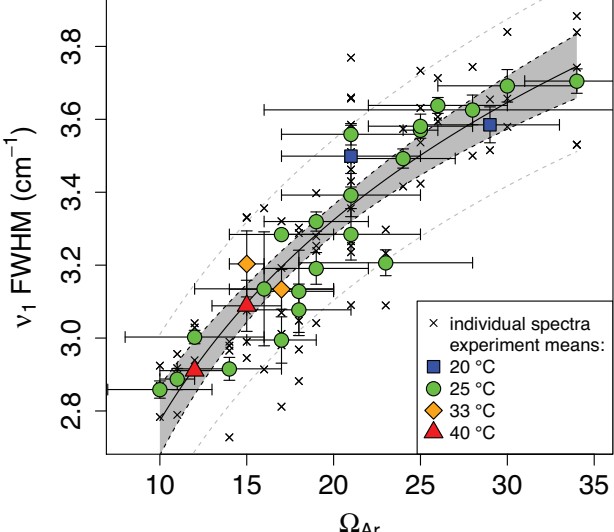

**Figure 2.** Sensitivity of true $\nu_1$ FWHM to $\Omega_{Ar}$. The x-axis represents the mean, and the horizontal error bars represent 1 standard deviation, of $\Omega_{Ar}$ while aragonite precipitated in each experiment. The crosses indicate individual spectra, the filled points indicate means of each experiment, and the vertical error bars represent the standard error of replicate spectra. Colours indicate experiments conducted at various temperatures. Grey shading represents the standard error of the curve fit to the individual spectra, and the dashed grey lines represent the standard error of prediction.

Some uncertainty exists in defining the $\Omega_{Ar}$ associated with each Raman spectrum, due to the variability of $\Omega_{Ar}$ over the course of each abiogenic experiment. We conducted a Monte Carlo simulation using the data from the Horiba instrument to evaluate the variability in $\nu_1$ FWHM that is expected to arise from the uncertainty in $\Omega_{Ar}$. In each of $10^4$ Monte Carlo iterations, we added random error to the mean $\Omega_{Ar}$ of each experiment, based on the standard deviation of measured $\Omega_{Ar}$ and assuming

a Gaussian distribution. Next, we calculated $\nu_1$ FWHM in each iteration based on $\Omega_{Ar}$ (including the random error added) using the equation in Table 2. The average standard deviation of the Monte Carlo $\nu_1$ FWHM residuals (the difference between the predicted and calculated $\nu_1$ FWHM in each iteration) was 0.24 cm$^{-1}$, which is 50% greater than the observed standard deviation of the residuals (0.16 cm$^{-1}$). Since the simulated variability exceeds the observed variability, one or more of the following is implied: 1) most (or all) of the scatter in $\nu_1$ FWHM around the regression line is explained by uncertainty of $\Omega_{Ar}$,

2) the Raman spectra integrate signal from multiple crystals that together approximate the mean $\Omega_{Ar}$ during each experiment, and/or 3) variability in $\Omega_{Ar}$ is overestimated. Regardless, the data are consistent with a tight, logarithmic dependence of $\nu_1$ FWHM on $\Omega_{Ar}$, with scatter in $\Omega_{Ar}$ adding uncertainty to the observed relationship.

Analysis of 440 spectra collected from various grains of JCp-1 produced a mean measured $\nu_1$ FWHM of 3.51 with $1\sigma$ of $\pm$ 0.09 cm$^{-1}$ (true $\nu_1$ FWHM of 2.99 $\pm$ 0.11 cm$^{-1}$). Applying the calibration equation in Table 2, this translates to a mean

derived $\Omega_{Ar}$ of 12.3, with a standard deviation of 2.1, and a standard error of the mean of 0.3.




**Table 1.** Linear model statistics for fits to $\nu_1$ FWHM

| Model: | F-value[a] | p-value(s) | $r^2$ | Residual $\sigma$[b] |
|---|---|---|---|---|
| $\ln(\Omega_{Ar})$ | 186.4 | **<0.001** | 0.70 | 0.165 |
| Mg/Ca | 85.79 | **<0.001** | 0.52 | 0.210 |
| G | 0.77 | 0.383 | 0.0 | 0.304 |
| $CO_3^{2-}$ | 133.9 | **<0.001** | 0.63 | 0.185 |
| pH | 35.46 | **<0.001** | 0.31 | 0.253 |
| $T$ | 6.75 | **0.011** | 0.07 | 0.293 |
| $\nu_1$ height | 4.50 | **0.037** | 0.04 | 0.297 |
| $\nu_1$ position | 0.06 | 0.807 | 0.02 | 0.289 |
| $\ln(\Omega_{Ar})$ + Mg/Ca | 92.97 | **<0.001**, 0.460 | 0.70 | 0.166 |
| $\ln(\Omega_{Ar})$ + G | 92.02 | **<0.001**, 0.944 | 0.70 | 0.166 |
| $\ln(\Omega_{Ar})$ + $CO_3^{2-}$ | 95.91 | **<0.001**, 0.135 | 0.71 | 0.164 |
| $\ln(\Omega_{Ar})$ + pH | 97.43 | **<0.001**, 0.079 | 0.71 | 0.163 |
| $\ln(\Omega_{Ar})$ + $T$ | 92.18 | **<0.001**, 0.755 | 0.70 | 0.166 |
| $\ln(\Omega_{Ar})$ + $\nu_1$ height | 92.02 | **<0.001**, 0.954 | 0.70 | 0.166 |

Notes: None of the multivariate models were significantly different from the $\Omega_{Ar}$ model. [a] 1,77 degrees of freedom for univariate models, and 2,76 degrees of freedom for multivariate models; except for $\nu_1$ position, which had 1,59 degrees of freedom. [b] standard deviation (1 $\sigma$) of the $\nu_1$ FWHM residuals from the model fit.

**Table 2.** Regression equations between mean $\nu_1$ FWHM and $\Omega_{Ar}$

| Calibration: | Intercept | Slope | $r^2$ | Residual $\sigma$ |
|---|---|---|---|---|
| Horiba measured | 1.62 (0.17) | 0.70 (0.06) | 0.85 | 0.09 |
| Horiba true | 0.94 (0.19) | 0.79 (0.06) | 0.85 | 0.11 |
| WITec measured | 2.09 (0.14) | 0.57 (0.05) | 0.96 | 0.05 |
| WITec true | 1.35 (0.16) | 0.66 (0.06) | 0.96 | 0.06 |

Notes: Equations describe $\nu_1$ FWHM = slope$\times\ln(\Omega_{Ar})$ + intercept, where the regressions were performed on the mean values of each experiment. Parentheses indicate 1 standard error.



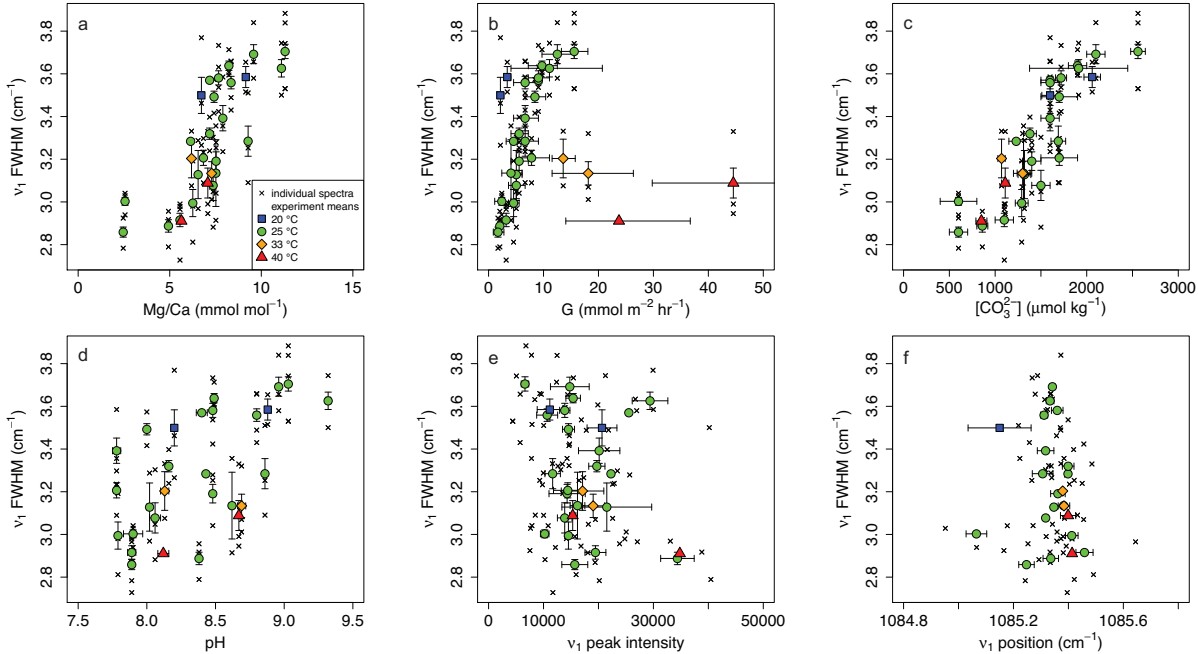

**Figure 3.** Scatter plots of Raman true peak width ($\nu_1$ FWHM) and (a) aragonite Mg/Ca, (b) aragonite precipitation rate, (c) experimental fluid [$CO_3^{2-}$], (d) experimental fluid pH, (e) Raman peak intensity, and (f) Raman peak position. While some apparent correlations exist, statistical models (Table 1) imply that $\Omega_{Ar}$ is the variable controlling $\nu_1$ FWHM, and that the patterns observed here are artefacts of correlations between these factors and $\Omega_{Ar}$.

In the Havannah Island coral, Raman-derived $\Omega_{Ar}$ showed clear annual cycles. The amplitude of annual cycles decreased from approximately 3 $\Omega_{Ar}$ units before the 1998 thermal stress event to approximately 1-2 $\Omega_{Ar}$ units afterwards.

Raman-derived $\Omega_{Ar}$ of the cultured *Acropora* was significantly different among treatments. Tukey's test revealed significant decreases in $\Omega_{Ar}$ as both $T$ (p < 0.001) and $CO_2$ (p = 0.045) increased. There was no significant interactive effect of $T$ and
5  $CO_2$.

$K_D^{\text{Mg/Ca}}$ determined from the abiogenic aragonites was significantly positively correlated with $\Omega_{Ar}$ (Fig. 4; Supplement Table S1; r$^2$ = 0.82 for experiments conducted at 25.5 °C):

$$K_D^{\text{Mg/Ca}} = \frac{(0.045 \pm 0.004)\Omega_{Ar} + 0.31 \pm 0.11}{1000} \tag{5}$$

The key finding of the $K_D^{\text{Mg/Ca}}$ data for this study is the sensitivity to $\Omega_{Ar}$ at a single temperature. This dependence of $K_D^{\text{Mg/Ca}}$
10  on $\Omega_{Ar}$ or precipitation rate was predicted by Gaetani and Cohen (2006) on the basis of the surface entrapment model proposed by Watson (2004), and was also observed in a recent abiogenic aragonite study (AlKhatib and Eisenhauer, 2017) (Fig. 4).



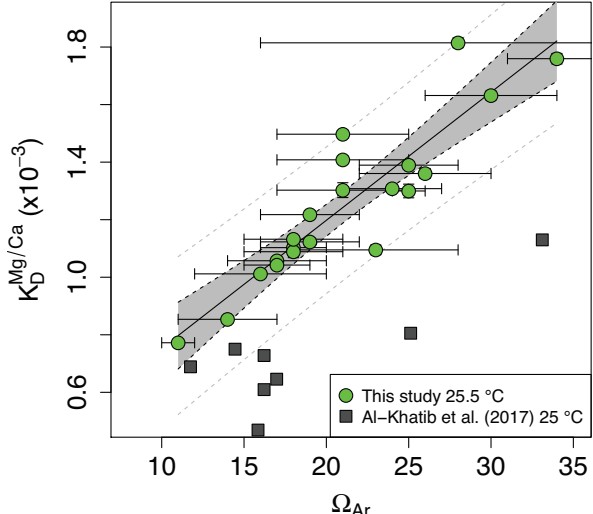

**Figure 4.** Sensitivity of $K_D^{\mathrm{Mg/Ca}}$ to $\Omega_{Ar}$ at 25 °C. Horizontal error bars represent 1 standard deviation of $\Omega_{Ar}$ while aragonite precipitated in each experiment. Grey shading represents the standard error of the curve, and the dashed grey lines represent the standard error of prediction. The grey squares show the AlKhatib and Eisenhauer (2017) data replotted against $\Omega_{Ar}$ (*cf* their Fig. 4 for a plot against precipitation rate).

## 4 Discussion

### 4.1 Controls on aragonite $\nu_1$ peak width

We conducted Raman spectroscopy analyses of abiogenic aragonites, precipitated under various carbonate chemistry and temperature treatments as previously described in DeCarlo et al. (2015) and Holcomb et al. (2016). Carbonate chemistry was

manipulated in these experiments such that there is some independence between the carbonate system parameters, which allows us to isolate their effects on the Raman spectral peaks. For example, experiments conducted at the same $\Omega_{Ar}$, but different temperatures, allow us to identify that $\nu_1$ FWHM is sensitive to $\Omega_{Ar}$ but not to aragonite precipitation rate ($G$) (Figs. 2 and 3b). Other factors were more difficult to isolate, for instance $\Omega_{Ar}$ was correlated with both solid Mg/Ca and fluid [$CO_3^{2-}$]. Yet, comparisons among statistical models (Table 1) show that $\Omega_{Ar}$ is the only variable that consistently has a significant effect

on $\nu_1$ FWHM, regardless of which additional variable is considered in multivariate models. This leads us to conclude that $\nu_1$ FWHM is primarily dependent upon $\Omega_{Ar}$. We also found that the scattering of data around the regression between $\nu_1$ FWHM and $\Omega_{Ar}$ can be explained entirely by the uncertainty of $\Omega_{Ar}$ in each experiment. Further, the absence of any significant correlations between $\nu_1$ FWHM and either $\nu_1$ position or height indicates that the variation of $\nu_1$ FWHM in our study was not the result of instrumental or curve-fitting artefacts. Together, our analyses demonstrate that Raman $\nu_1$ FWHM is a reliable proxy

for the $\Omega_{Ar}$ of the fluid from which aragonite precipitated.





The influence of $\Omega_{Ar}$ on $\nu_1$ FWHM can be explained by carbonate ion disorder within the aragonite lattice, as originally hypothesized by Bischoff et al. (1985). Increasing $\Omega_{Ar}$ causes more defects and/or impurities to be incorporated into the aragonite, increasing the positional disorder of carbonate in the lattice. Disordered carbonate ions have a distribution of C-O bond strengths and vibrational frequencies, which results in wider Raman peaks (Bischoff et al., 1985; Urmos et al., 1991;

Perrin et al., 2016).

Raman spectroscopy studies of abiogenic $CaCO_3$ have so far focused on calcite, vaterite, and amorphous $CaCO_3$ (ACC) precipitated with wide ranges of Mg/Ca (from 0-45 mol%) (Bischoff et al., 1985; Wehrmeister et al., 2009; Wang et al., 2012; Perrin et al., 2016). These studies manipulated the Mg/Ca ratios of the solutions from which the precipitates formed, and they consistently reported strong correlations between $\nu_1$ FWHM and $\nu_1$ position, both of which increased with progressively higher

Mg/Ca in the solid phase. The abiogenic precipitates used in our study are different in that (1) they are aragonite, and (2) most of them (26 out of 28) were grown from the same parent solution (seawater) but under different carbonate chemistry treatments (DeCarlo et al., 2015; Holcomb et al., 2016). Most of the Mg/Ca variability in these aragonites arises from the sensitivity of Mg/Ca partitioning to $\Omega_{Ar}$ (Fig. 4), not manipulation of Mg/Ca in the initial fluid. Recognising this methodological distinction is important for understanding why we did not find a correlation between $\nu_1$ FWHM and position. Firstly, aragonite has several

orders of magnitude less Mg than calcite, potentially dampening the role of Mg in affecting the shape of the Raman peaks in aragonite. Secondly, if the trend between $\Omega_{Ar}$ and $\nu_1$ FWHM were driven by Mg content alone, shifts in Raman peaks to higher wavenumbers would be expected because Mg is lighter than Ca (Bischoff et al., 1985). However, there was no significant correlation between $\nu_1$ position and $\Omega_{Ar}$ (p > 0.7), indicating that the mean C-O bond vibrational frequency did not change systematically with $\Omega_{Ar}$. These observations imply that Mg content was not the primary source of lattice defects broadening

Raman peaks, an interpretation supported by the absence of a correlation between Mg/Ca and $\nu_1$ FWHM (after accounting for the effect of $\Omega_{Ar}$; Table 1). The lack of correlation between $\nu_1$ FWHM and position also indicates that $\nu_1$ FWHM was unaffected by crystallite size (Urmos et al., 1991; Zakaria et al., 2008).

Our finding that fluid $\Omega_{Ar}$, not solid Mg/Ca, controls $\nu_1$ FWHM in aragonite precipitated from seawater is critical for interpreting Raman spectra of coral skeletons. Corals transport seawater to the micro-scale site of calcification (Gagnon et al.,

2012; Tambutté et al., 2012), and much like in the abiogenic experiments of DeCarlo et al. (2015) and Holcomb et al. (2016), they precipitate aragonite crystals from a seawater-like solution (McConnaughey, 1989). Changes in calcifying fluid Mg/Ca can result from $Ca^{2+}$ transport and/or Rayleigh fractionation (Gaetani and Cohen, 2006; Gagnon et al., 2012). Yet Mg/Ca in coral skeletons is generally within the limits of the abiogenic aragonites analysed here (2.5 - 11 mmol mol$^{-1}$), and critically, $\nu_1$ FWHM was insensitive to Mg/Ca over this range. This implies that we can interpret Raman spectra of coral skeletons based

on the abiogenic sensitivity of $\nu_1$ FWHM to $\Omega_{Ar}$.

### 4.2    Analysis of JCp-1 and relation to skeletal geochemistry

Using Raman spectroscopy, we derived a mean calcifying fluid $\Omega_{Ar}$ for the JCp-1 coral standard of $12.3 \pm 0.3$. JCp-1 is an internationally calibrated geochemical standard, and has been analysed repeatedly for trace element and isotopic composition. Because the geochemistry of the skeleton reflects the calcifying fluid conditions, we can evaluate the accuracy of our $\Omega_{Ar}$



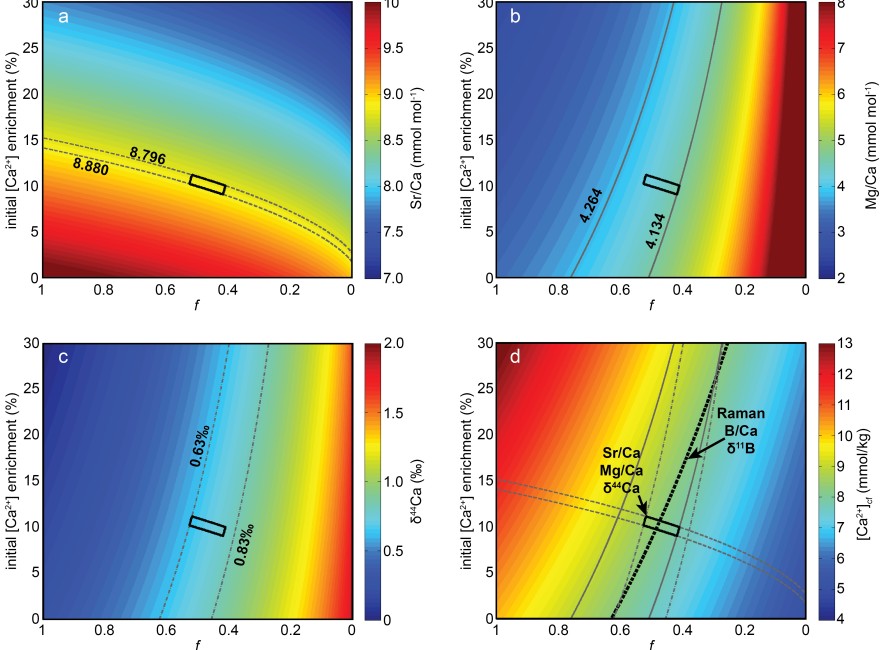

**Figure 5.** Evaluation of Raman-derived $\Omega_{Ar}$ based on geochemistry of the JCp-1 coral. In each panel, the x-axis $f$ is fraction of total $[Ca^{2+}]_{cf}$ remaining, and the y-axis is initial $[Ca^{2+}]_{cf}$ enrichment relative to seawater. Colours indicate calculated (a) skeletal Sr/Ca, (b) skeletal Mg/Ca, (c) skeletal $\delta^{44}$Ca, and (d) $[Ca^{2+}]_{cf}$. In (a-c) the grey lines bound the reported values of JCp-1 ($\pm 1\sigma$), and the black polygon shows the parameter space consistent with Sr/Ca, Mg/Ca, and $\delta^{44}$Ca. Panel (d) shows $[Ca^{2+}]_{cf}$ and the grey lines show the constraints from panels (a-c). The dashed black line indicates the $[Ca^{2+}]_{cf}$ independently estimated based on combining our Raman data with published boron data (see text for details of the calculations). The two approaches are within error (*i.e.* the dashed black line intersects the solid black polygon) for Raman-derived $\Omega_{Ar}$ between 10.8 and 13.7, consistent with the observed $\Omega_{Ar}$ of $12.3 \pm 0.3$.

estimates by comparing them to the reported geochemical properties of the JCp-1 carbonate. Specifically, here we calculate calcifying fluid calcium concentration ($[Ca^{2+}]_{cf}$) using two different sets of proxies; the first based on Sr/Ca, Mg/Ca and $\delta^{44}$Ca, and the second based on Raman, $\delta^{11}$B (a pH proxy), and B/Ca (see Appendix A for details of calculations). Our approach uses experimentally determined partitioning of these trace elements and isotopes between aragonite and seawater, and a simple model in which $[Ca^{2+}]_{cf}$ is initially elevated relative to seawater (Al-Horani et al., 2003) before precipitation

5  from the isolated calcifying fluid (*i.e.* a closed system) drives Rayleigh fractionation (Gaetani and Cohen, 2006; Gaetani et al., 2011; Gagnon et al., 2012). We then compare these two independent $[Ca^{2+}]_{cf}$ estimates to test whether our Raman results are consistent with JCp-1 geochemistry.

The published Mg/Ca, Sr/Ca and $\delta^{44}$Ca of the JCp-1 coral can be explained with initial $Ca^{2+}$ enrichment of 9-11% and

10  precipitation of 48-59% of the total $Ca^{2+}$. This corresponds to mean $[Ca^{2+}]_{cf}$ of $8.2 \pm 0.7$ mmol kg$^{-1}$ (Fig. 5). We then independently estimated $[Ca^{2+}]_{cf}$ by combining our Raman spectroscopy results with boron systematics. The $\delta^{11}$B and B/Ca



of JCp-1 together imply a calcifying fluid $[CO_3^{2-}]$ of $987 \pm 78$ $\mu$mol kg$^{-1}$. To reconcile the boron-derived $[CO_3^{2-}]$ with the Raman-derived $\Omega_{Ar}$ of $12.3 \pm 0.3$ requires $[Ca^{2+}]_{cf}$ of $8.3 \pm 0.7$ mmol kg$^{-1}$. This agrees almost exactly with the mean $[Ca^{2+}]_{cf}$ based on the calculations described above for Sr/Ca, Mg/Ca, and $\delta^{44}$Ca systematics (Fig. 5d), giving us confidence in the accuracy of our Raman-based $\Omega_{Ar}$ results. One implication of these results is that $[Ca^{2+}]_{cf}$ is less than seawater by

11-25%, and while this finding should be replicated on additional corals, we note that depleted $[Ca^{2+}]_{cf}$ is consistent with trace element variability in deep-sea and surface-dwelling corals modelled using both "batch" and "flow-through" versions of the Rayleigh calculations (Gaetani et al., 2011; Gagnon et al., 2012). Further, our Raman-derived $\Omega_{Ar}$ is generally consistent with estimates made on the basis of coral skeleton crystal aspect ratios (Cohen and Holcomb, 2009).

### 4.3   High-resolution time series of calcifying fluid $\Omega_{Ar}$

Analysis of the Havannah Island coral demonstrates that Raman spectroscopy can detect variability in coral calcifying fluid $\Omega_{Ar}$. The pattern of variability in Raman-derived $\Omega_{Ar}$ showed some similarities, but also some differences, compared to the boron-derived $\Omega_{Ar}$ presented in D'Olivo and McCulloch (2017). In a sampling track that avoided a 1998 partial-mortality scar ("Path D"), boron-derived $\Omega_{Ar}$ showed annual oscillations of 2-3 units, but one seasonal oscillation was completely missing during 1998/1999 (Fig. 6). Yet in a track directly adjacent to the scar ("Path B"), boron-derived $\Omega_{Ar}$ showed overall lower

values and less variability, compared to that from Path D. Raman-derived $\Omega_{Ar}$ closely tracked variability of Path D from 1996 to early 1998, then diverged from the boron-derived $\Omega_{Ar}$ between mid-1998 and mid-1999, before returning to similar seasonal variability by 2000.

There are several possible reasons for the differences between Raman- and boron- derived $\Omega_{Ar}$. The first is that the Raman sampling path was located approximately halfway between boron Paths B and D, and thus there could be differences in

$\Omega_{Ar}$ across the skeleton. The boron data indicate lower $\Omega_{Ar}$ near the partial-mortality scar while the Raman data potentially reflect an intermediate level of bleaching stress between the low values of Path B and the higher values of Path D. There is also potential for the existence of secondary precipitates near the scar, although the Raman spectra clearly indicated only aragonite was present with no sign of calcite. Second, the Raman measurements have higher spatial resolution than the boron measurements, potentially allowing us to capture variability not observed in the boron sampling paths. Finally, boron-derived

estimates of $\Omega_{Ar}$ are based on the assumption that $[Ca^{2+}]_{cf}$ remains at, or close to, seawater concentrations, and therefore any substantial changes in $[Ca^{2+}]_{cf}$ will not be recorded in the boron-derived time series. This means that the decoupling between Raman- and boron-derived $\Omega_{Ar}$ from Path D during 1998-1999 could reflect changes in $[Ca^{2+}]_{cf}$ (Fig. 6). D'Olivo and McCulloch (2017) showed that trace element ratios (Mg/Ca and Sr/Ca) in Path B indicated a strong reduction in $[Ca^{2+}]_{cf}$ during 1998, although a similar pattern was not observed in Path D. Similarly, a recent study showed that Sr/Ca and Mg/Ca

responses to bleaching in *Porites* from Western Australia were indicative of reduced $[Ca^{2+}]_{cf}$ (Clarke et al., 2017), consistent with our comparison between Raman and boron systematics in the Havannah Island coral. Mechanistically, reduced $[Ca^{2+}]_{cf}$ could result from decreases in the renewal rates of calcifying fluid (via increased Rayleigh fractionation) or decreases in Ca$^{2+}$ addition to the calcifying fluid. While more studies are needed to test if reduced $[Ca^{2+}]_{cf}$ is a consistent response to thermal stress, the ability to detect such changes by using Raman and boron systematics in tandem highlights their combined utility.




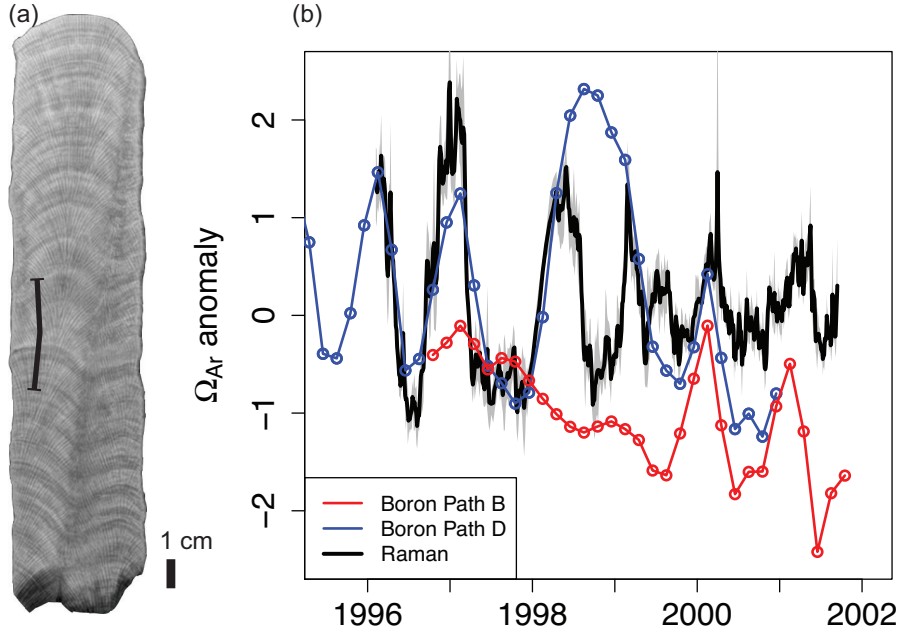

**Figure 6.** Time series of boron- and Raman-derived $\Omega_{Ar}$ in a coral from Havannah Island. (a) X-Ray image of central slab cut from the core. The black line indicates the Raman sampling path. Boron paths B and D were located approximately 1 cm into and out of the page, respectively (*cf.* Fig. 1 in D'Olivo and McCulloch (2017) for details). (b) The red and blue lines show boron-derived $\Omega_{Ar}$ assuming a constant $[Ca^{2+}]_{cf}$. Raman-derived $\Omega_{Ar}$ with grey error bars indicating 1 $\sigma$. The means of the two datasets (mean of Raman $\Omega_{Ar}$ was 9.5) have been removed to facilitate comparison and because of uncertainty of absolute $[Ca^{2+}]_{cf}$ for the boron calculations.

## 4.4 Effects of temperature and CO$_2$ on $\Omega_{Ar}$ of juvenile *Acropora*

Our analyses of juvenile *Acropora* indicate that calcifying fluid $\Omega_{Ar}$ decreased in response to elevated temperature (3 °C above the maximum monthly mean water temperature) and to elevated CO$_2$ (Fig. 7). In micro-computed tomography (micro-CT) analyses of corals from the same experiment, Foster et al. (2016) showed that elevated CO$_2$ increased skeletal porosity

5   and decreased total calcification, while elevated temperature had the effect of partially mitigating the response to elevated CO$_2$. Thus, the effect of elevated CO$_2$ on Raman-derived $\Omega_{Ar}$ is consistent with the calcification response, but thermal stress decreased $\Omega_{Ar}$ without having a negative impact on calcification. One possibility is that the effect of decreasing $\Omega_{Ar}$ on aragonite accretion was balanced by elevated temperature because aragonite crystal growth rates increase with both $\Omega_{Ar}$ and temperature (Burton and Walter, 1987). Further, the physiological response to elevated temperature could be complex,

10   potentially involving decreases in $\Omega_{Ar}$ balanced by increases in calcifying time or surface area. Finally, Foster et al. (2016) found pitted skeletal surfaces indicative of dissolution, which may have compounded the measured CO$_2$ effects on calcification and led to their dominance over any apparent effect of temperature in the "+CO$_2$+*T*" treatment. A recent study employing a similar experimental design found that juvenile *Acropora* B/Ca and U/Ca increased with elevated CO$_2$ (Wu et al., 2017),





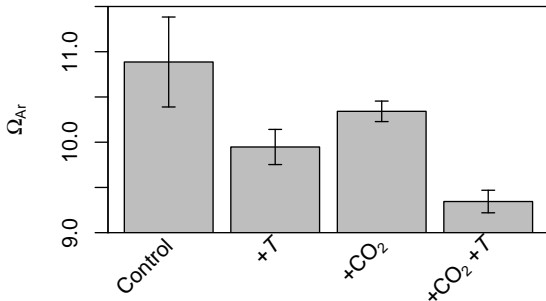

**Figure 7.** Raman-derived $\Omega_{Ar}$ of *Acropora* cultured under different temperature and $CO_2$ treatments. The grey bars indicate treatment means and error bars represent $\pm$ standard error of the mean. Significant effects were observed for both temperature and $CO_2$.

both of which indicate decreases in calcifying fluid $[CO_3^{2-}]$ based on abiogenic partitioning (DeCarlo et al., 2015; Holcomb et al., 2016). However, while elevated temperature increased $\delta^{11}B$ it did not affect B/Ca or U/Ca (Wu et al., 2017), potentially reflecting the lower degree of thermal stress (culture temperatures were within the natural seasonal range at the collection site) compared to Foster et al. (2015) and/or that $[Ca^{2+}]_{cf}$ played an important role in our Raman data. Nevertheless, our
results further highlight that Raman spectroscopy can detect changes in coral calcifying fluid $\Omega_{Ar}$, and that this information is complementary to other analyses, including boron systematics (section 4.3) as well as CT-derived porosity and calcification.

## 5 Conclusions and Outlook

We tested whether Raman spectroscopy records the saturation state of the fluid from which aragonite precipitates. In abiogenic aragonites with known fluid conditions, we found a clear dependence of Raman peak width on $\Omega_{Ar}$, an observation that is not
confounded by other factors, including temperature and Mg/Ca. In the JCp-1 coral standard, for which calcifying fluid $\Omega_{Ar}$ is not known directly, Raman-derived $\Omega_{Ar}$ is consistent with multiple, independent lines of geochemical evidence based on Sr/Ca, Mg/Ca, B/Ca, $\delta^{44}Ca$, and $\delta^{11}B$ ratios. Annual cycles of Raman-derived $\Omega_{Ar}$ in a field-collected *Porites* and $\Omega_{Ar}$ responses to elevated temperature and $CO_2$ in cultured *Acropora* suggest that Raman spectroscopy is indeed capable of detecting changes in coral calcifying fluid $\Omega_{Ar}$.
Our approach could potentially be applied to other marine calcifiers that build aragonitic shells and skeletons, such as sclerosponges and some molluscs. However, the relationship between Raman peak width and $\Omega$ is likely to be mineral-specific. For example, calcite often contains magnesium with concentrations several orders of magnitude greater than in aragonite, which is likely the reason for the wider peaks in Raman spectra of calcite (Urmos et al., 1991). In contrast, we have shown





that Raman spectroscopy is well suited for analysis of coral calcifying fluid $\Omega_{Ar}$. Additionally, the high spatial resolution (< 1 $\mu$m) and rapid sample processing ($\leq$1 s per spectrum) of Raman are currently unrivalled. Further, the same Raman spectra used to estimate $\Omega_{Ar}$ can confirm that the sample is aragonite, effectively eliminating the possibility of contamination from other mineral phases. Finally, our analysis indicates that the Raman $\nu_1$ FWHM is an accurate proxy of fluid $\Omega_{Ar}$, making it a complementary approach to B/Ca and $\delta^{11}$B because combining information from the two approaches allows calculating the full carbonate system via [CO$_3^{2-}$], pH and now importantly [Ca$^{2+}$]$_{cf}$.

The $\Omega_{Ar}$ of 12.3 that we derived for JCp-1 corresponds to an instantaneous inorganic (Burton and Walter, 1987) aragonite precipitation rate of 0.2 g cm$^{-2}$ yr$^{-1}$. However, the average annual calcification rate of *Porites* living at 25 °C is typically 1.3 g cm$^{-2}$ yr$^{-1}$ (Lough, 2008), a factor of x6.5 faster than expected on the basis of the inorganic calcifying fluid $\Omega_{Ar}$ systematics. Since coral calyces have complex 3-dimensional shape such that their skeletal surface area is actually much greater than the planar area of the colony surface (Barnes, 1970), this implies that calcifying surface area is an important factor in determining coral calcification rates (D'Olivo and McCulloch, 2017) and it complicates the use of bulk calcification rates for estimating $\Omega_{Ar}$ (Raybaud et al., 2017). Multiple studies using micro-CT have shown increases in porosity and surface area:volume ratios in corals cultured under elevated CO$_2$ levels (Tambutté et al., 2015; Foster et al., 2016). This may represent a strategy for controlling calcifying surface area (*i.e.* the area over which crystals are precipitating from the calcifying fluid at a given time) to maintain bulk calcification rates and partially offset the impacts of ocean acidification.

In summary, recent studies and technological advances in boron isotope measurements (McCulloch et al., 2014), CT (De-Carlo and Cohen, 2016; Foster et al., 2016) and Raman spectroscopy (this study) are now making it increasingly feasible to quantify coral calcification responses with a multi-pronged approach. Future investigations combining Raman spectroscopy with B/Ca, $\delta^{11}$B, and porosity measurements on corals grown under elevated CO$_2$ will be valuable for understanding mechanisms of resilience to ocean acidification. These include pH homeostasis, maintenance of high calcifying fluid $\Omega_{Ar}$, and increases in calcifying surface area as potential strategies for coral calcification to persist in a high-CO$_2$ world. Evaluating their relative importance in the face of both increasing reef-water temperature and declining [CO$_3^{2-}$] will be critical to informing our predictions of where and when resilience may be found.

## 6 Code availability

An R code for Raman curve fitting is included in the Supplement. Codes for the full analysis will be posted on Code Ocean.

## 7 Data availability

Raman data are summarised in the Supplement. Individual Raman spectra will be posted on Zenodo and Code Ocean.





## Appendix A: Details of JCp-1 geochemistry calculations

The JCp-1 coral grew at an average temperature of 25 °C (Okai et al., 2002), and the mean salinity in the area is 34.54 (Levitus, 2010). At 25 °C, the $K_D^{Sr/Ca}$ is 1.123 (DeCarlo et al., 2015), and at 25 °C and $\Omega_{Ar}$ of 12, $K_D^{Mg/Ca}$ is ~6.5($\pm$0.5)x10$^{-4}$ (Fig. 4; in agreement with the mean value of AlKhatib and Eisenhauer (2017) at < 20 $\Omega_{Ar}$ and within error of our regression). We

calculated a range of possible skeletal Sr/Ca and Mg/Ca ratios based on various combinations of Ca$^{2+}$ pumping (enrichment of initial $[Ca^{2+}]_{cf}$ relative to seawater between 0 and 30%) and Rayleigh fractionation (0-100% of total Ca$^{2+}$ precipitated), and we compared the results to the known Sr/Ca (8.838 $\pm$ 0.042 mmol mol$^{-1}$) and Mg/Ca (4.199 $\pm$ 0.065 mmol mol$^{-1}$) ratios of JCp-1 (Fig. 5) (Hathorne et al., 2013). For $\delta^{44}$Ca, we used the abiogenic aragonite fractionation factor (Gussone et al., 2003, 2005), seawater $\delta^{44}$Ca = 1.88‰ (Hippler et al., 2003), Ca added via Ca-ATPase (*i.e.* $[Ca^{2+}]_{cf}$ enrichment) of 0.68‰ (Inoue

et al., 2015), and an average coral skeleton $\delta^{44}$Ca of 0.73 $\pm$ 0.1 (Inoue et al., 2015; Chen et al., 2016; Gothmann et al., 2016). Our calculation scheme utilised the Rayleigh equation:

$$\frac{i}{j}_{coral} = \frac{i}{j}_{initial\ fluid} \frac{(1 - f^D)}{1 - f} \tag{A1}$$

where $i/j$ is Sr/Ca, Mg/Ca or $^{44}$Ca/$^{40}$Ca, $f$ is the fraction of Ca remaining, $D$ is $K_D^{Sr/Ca}$, $K_D^{Mg/Ca}$ or $\alpha_{44-40}$, and it is assumed that Sr and Mg concentrations in the initial fluid prior to precipitation are equal to those of seawater.

A similar result is achieved when considering the calcifying fluid as semi-closed and using the steady-state "flow-through" model of Gagnon et al. (2012). We used the same $K_D^{Sr/Ca}$ and $K_D^{Mg/Ca}$ listed above and the equations of Gagnon et al. (2012) to solve for their "$\gamma$" term (defined as the ratio of calcium pumping to precipitation) that best fit the JCp-1 data (0.35). Next, we found the unique value of "$\frac{P}{kz\rho}$" (where $P$ is precipitation flux, $k$ is seawater exchange rate, $z$ is the ratio of calcifying volume to surface area, and $\rho$ is seawater density) that was consistent with this $\gamma$ and the JCp-1 Sr/Ca and Mg/Ca (3.1 mmol Ca kg$^{-1}$).

Finally, using these terms matched to JCp-1 geochemistry, we follow the equation given in Gagnon et al. (2012):

$$[Ca^{2+}]_{cf} = [Ca^{2+}]_{seawater} - \frac{(1 - \gamma)P}{kz\rho} \tag{A2}$$

The $[Ca^{2+}]_{cf}$ calculated by this method is 8.1 mmol kg$^{-1}$, within error of the 8.2 $\pm$ 0.7 mmol kg$^{-1}$ derived from our "batch" model calculations above (see Fig. 5).

Our second approach to calculating $[Ca^{2+}]_{cf}$ combines Raman spectroscopy with boron systematics. JCp-1 has a B/Ca

ratio of 459.6 $\pm$ 22.7 $\mu$mol/mol (1$\sigma$) (Hathorne et al., 2013) and $\delta^{11}$B of 24.3 $\pm$ 0.17‰ (1$\sigma$) (McCulloch et al., 2014). We used the isotope fractionation factor of Klochko et al. (2006) to determine pH from $\delta^{11}$B assuming seawater $\delta^{11}$B of 39.6‰ (Foster et al., 2010). Next, we used the parameterisation of pK$_B$ from Dickson (1990) to calculate [B(OH)$_4^-$] from pH, temperature, salinity, and assuming [B] equal to seawater (411 $\mu$mol kg$^{-1}$ at salinity 34.54) (Allison et al., 2014). The $K_D^{B/Ca}$ of Holcomb et al. (2016) was used to determine [CO$_3^{2-}$] from measured B/Ca and calculated [B(OH)$_4^-$] (computer code

available at https://codeocean.com/2017/05/08/boron-systematics-of-aragonite/interface, and see McCulloch et al. (2017)). We then solve for $[Ca^{2+}]_{cf}$ with the following formula:

$$[Ca^{2+}]_{cf} = \frac{\Omega_{Ar}}{[CO_3^{2-}]} K_{sp} \tag{A3}$$




where $\Omega_{Ar}$ is derived from Raman spectroscopy and $K_{sp}$ is the solubility product of aragonite in seawater. $[Ca^{2+}]_{cf}$ calculated in this way is $8.3 \pm 0.7$ mmol kg$^{-1}$.

In the first set of calculations with Sr/Ca, Mg/Ca and $\delta^{44}$Ca, we assume only that Ca$^{2+}$ is added to the fluid, whereas in the second set of calculations with Raman spectroscopy and boron systematics we assume carbonate chemistry is modified

from seawater. However, since we do not directly connect the initial Ca$^{2+}$ enrichment of the fluid with the carbonate system calculations, we do not need to make assumptions regarding the stoichiometry of any potential proton-Ca$^{2+}$ exchange.

*Author contributions.*  T.M.D. conceived the idea, designed the study, conducted Raman measurements, and analysed the data. M.H., J.P.D., T.F., and M.T.M. contributed materials and aided in the interpretation of results. T.M.D. wrote the manuscript and all authors contributed to revising the final version.

*Competing interests.*  The authors declare no competing interests.

*Acknowledgements.*  We thank all those who contributed to the materials used in this study. The abiogenic aragonites were precipitated in the laboratories of Dr. Glenn Gaetani and Dr. Anne Cohen at Woods Hole Oceanographic Institution (WHOI). For the study of the GBR coral, H. Clarke and K. Rankenburg at University of Western Australia provided laboratory and analytical assistance. Facilities and technical assistance for the *Acropora* culturing experiments were provided by the Batavia Coast Maritime Institute, with logistical and technical

support from A. Basile and T. Basile. Frieder Klein (WHOI) assisted with the initial Raman measurements. Funding was provided by an ARC Laureate Fellowship (FL120100049) awarded to Professor Malcolm McCulloch and the ARC Centre of Excellence for Coral Reef Studies (CE140100020). The authors acknowledge the facilities, and the scientific and technical assistance of the Australian Microscopy & Microanalysis Research Facility at the Centre for Microscopy, Characterisation & Analysis, The University of Western Australia, a facility funded by the University, the Western Australian State and Commonwealth Governments.





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
