# Peer review of "Coral calcifying fluid aragonite saturation states derived from Raman spectroscopy"

_Biogeosciences, 2017_

## Referee Comment (RC1) · J. Fietzke (Referee) · 23 Jul 2017

The submitted manuscript of DeCarlo et al. reports interesting innovative research results regarding the use of Raman spectroscopy for the determination of aragonite saturation state in inorganic experiments and during coral calcification. As such it is timely, of broad scientific interest and fits into the scope of BG.

I would expect this material to be publishable after careful revision.

Before explaining some of my concerns I need to underline I am not at all a Raman spectroscopy expert. Thus, Raman-specific technical details need to be reviewed by a respective expert before any decision on publication can be made.

In general I like this manuscript for it's interesting approach which warrents publication

in my opinion. Nevertheless, some conclusions, I think, should be presented more carefully, underlining the potential and open questions of this approach.

It appears strange to me, the calibration (inorganic) yields results for the coral which are presented as correct reconstruction of internal cf saturation state. The calibration in my opinion is not entirely correct or it is not very clearly explained. I'd tried to recalculate the regression based on the data provided in the supplements. It appears, the uncertainties of the saturation state data from the experiment have not been included in the uncertainty estimate of the calibration. It rather looks like the means of FWHM plotted vs. Omega and a log-fit applied. This is critical as later the FWHM is used to reconstruct Omega. I hope, the point is clear, it would need quite the opposite plot and fit, Omega vs. FWHM for a calibartion useful for the desired calculation. Well, the fit obviousely, changes in this case.

In any way, I could not replicate the Jcp-1 Omega of 12.3 with either of the calibrations. Could it be, each individual Raman result had been converted into a result for Omega and those results have been used to calculate an average of 12.3? If so, I did not get this from the manuscript... The Omega calculated from the mean FWHM would be >13, if I did the calculations right.

Considerring the large uncertainties of the source data (Omega from inorganic experiments), and the fact, that corals do not perform such experiments to grow their skeleton, it appears not realistc to claim the precise reconstruction of coral cf Omega +-1-2, as stated in the abstract.

Despite this critique I am confident and look forward to see this published as a paper in BG.

Cheers, Jan

---

## Author Comment (AC1) · 28 Jul 2017

*Reviewer comment 1*: The submitted manuscript of DeCarlo et al. reports interesting innovative research results regarding the use of Raman spectroscopy for the determination of aragonite saturation state in inorganic experiments and during coral calcification. As such it is timely, of broad scientific interest and fits into the scope of BG.

I would expect this material to be publishable after careful revision.

Before explaining some of my concerns I need to underline I am not at all a Raman spectroscopy expert. Thus, Raman-specific technical details need to be reviewed by a respective expert before any decision on publication can be made.

In general I like this manuscript for it's interesting approach which warrents publication

in my opinion. Nevertheless, some conclusions, I think, should be presented more carefully, underlining the potential and open questions of this approach.

*Response 1*: **We thank the reviewer for the supportive comments and careful consideration of our manuscript. We appreciate the issues raised, which highlight a couple areas of the manuscript that require clarification. Below, we respond to the reviewer's specific comments. While our results and conclusions remain the same, we agree with the reviewer that we need to add a few details to further explain how the calculations are performed.**

*Reviewer comment 2*: It appears strange to me, the calibration (inorganic) yields results for the coral which are presented as correct reconstruction of internal cf saturation state. The calibration in my opinion is not entirely correct or it is not very clearly explained. I'd tried to recalculate the regression based on the data provided in the supplements. It appears, the uncertainties of the saturation state data from the experiment have not been included in the uncertainty estimate of the calibration. It rather looks like the means of FWHM plotted vs. Omega and a log-fit applied. This is critical as later the FWHM is used to reconstruct Omega. I hope, the point is clear, it would need quite the opposite plot and fit, Omega vs. FWHM for a calibartion useful for the desired calculation. Well, the fit obviousely, changes in this case.

*Response 2*: **We appreciate the reviewer's comment, which has alerted us to a point that we will clarify in the revised manuscript. The reviewer is correct that our calibration is based on fitting FWHM to the log of $\Omega_{Ar}$ (see Table 2). We use $\Omega_{Ar}$ as the independent variable (x-axis in Figure 2) and FWHM as the dependent variable (y-axis in Figure 2) because our proposed mechanism is that $\Omega_{Ar}$ causes changes in FWHM (see section 4.1). However, we agree with the reviewer that this type of regression does not take errors in $\Omega_{Ar}$ into account, and that an alternative approach would be to fit $\Omega_{Ar}$ as a function of FWHM. We tried this method using the measured data from the WITec instrument (Figure R1-1 below). The difference between the two approaches is very small, e.g. approximately 0.1**

$\Omega_{Ar}$ **unit difference for JCp-1. Nevertheless, we will add a similar plot and the alternative calibration equation in the supplement for clarity.**

*Reviewer comment 3*: In any way, I could not replicate the Jcp-1 Omega of 12.3 with either of the calibrations. Could it be, each individual Raman result had been converted into a result for Omega and those results have been used to calculate an average of 12.3? If so, I did not get this from the manuscript... The Omega calculated from the mean FWHM would be >13, if I did the calculations right.

*Response 3*: **The reviewer is correct that we converted FWHM to $\Omega_{Ar}$ for each of the 440 JCp-1 measurements, and then we took the average of these $\Omega_{Ar}$ values. This will be clarified in the revised manuscript. However, the effect on the derived mean $\Omega_{Ar}$ is very small. We repeated these calculations using the reported JCp-1 mean FWHM (3.51 cm$^{-1}$) and the equation in Table 2 where $\Omega_{Ar}$ = $e^{\frac{(FWHM-2.09)}{0.57}}$, which gives 12.08. Thus, the difference from that reported in the manuscript (where $\Omega_{Ar}$ is calculated separately from each Raman spectra) is only 0.22 units. This is shown in the attached .R script, which also reproduces Figure R1-1 and calculates the JCp-1 derived $\Omega_{Ar}$ using the alternative calibration technique described in response to comment 2. In any case, the derived $\Omega_{Ar}$ is always between 12.08 and 12.30, which is within error of that reported in the manuscript (12.3 $\pm$ 0.3). Thus, while we cannot reproduce the $\Omega_{Ar}$ of 13 stated by the reviewer, we agree that these calculations were not described clearly enough in the manuscript. At the revision stage, we will explain the calculations in more detail and clarify that the JCp-1 analyses were conducted on the WITec instrument, as this may be the source of the confusion.**

**We also wish to point out that we have posted the .R scripts to repeat all the analyses in the manuscript and reproduce the figures at the following link: https://codeocean.com/2017/06/30/code-for-quot-coral-calcifying-fluid-aragonite-saturation-states-derived-from-raman-spectroscopy-quot/code**

**The calculations that produce JCp-1 $\Omega_{Ar}$ of 12.3 $\pm$ 0.3 from the raw data can be found there as well.**

*Reviewer comment 4*: Considerring the large uncertainties of the source data (Omega from inorganic ex- periments), and the fact, that corals do not perform such experiments to grow their skeleton, it appears not realistc to claim the precise reconstruction of coral cf Omega +-1-2, as stated in the abstract. Despite this critique I am confident and look forward to see this published as a paper in BG.

*Response 4*: **We agree with the reviewer that there are substantial uncertainties in $\Omega_{Ar}$ over the course of each abiogenic experiment, and this is discussed in detail on page 8. While the statistics of the calibration support our claim that $\Omega_{Ar}$ can be derived within 1-2 units, we will modify this statement as suggested by the reviewer to reflect that the calibration is developed on abiogenic aragonites. However, the comparison between coral geochemistry and our Raman-derived $\Omega_{Ar}$ supports the notion that our calibration does represent calcifying fluid $\Omega_{Ar}$.**

*Figure Caption*: **Figure R1-1. Measured Raman FWHM plotted against fluid $\Omega_{Ar}$ for the abiogenic aragonites analysed with the WITec instrument. The red curve shows the calibration when FWHM is fit as a function of $\Omega_{Ar}$, and the blue curve shows the calibration when $\Omega_{Ar}$ is fit as a function of FWHM. A similar plot with repeat measurements of this calibration over multiple days is shown in Figure S3.**

Please also note the supplement to this comment:
https://www.biogeosciences-discuss.net/bg-2017-194/bg-2017-194-AC1-supplement.zip

[Figure]

**Fig. 1.** Figure R1-1

---

## Referee Comment (RC2) · Anonymous Referee #2 · 15 Sep 2017

Dear editor,

I have carefully read the manuscript by DeCarlo and others that describes the use of Raman spectroscopy to determine conditions at which (biogenic) calcium carbonate is precipitated. Overall, this manuscript contains an impressive amount of data, including results from inorganic experiments, culture experiment and field data. I have one serious concern about the (in)directness of the relation between $\upsilon 1$ FWHM and the saturation state, that needs revising before this manuscript can be accepted for publication in Biogeosciences.

The correlation between the Raman shift and the $\Omega$ may well be specific to the (inorganic) experiment and may not be general. As the authors explain in the introduction and methods, the actual shift in the spectrum is caused by other elements (impurities)

[Figure]

or lattice distortions. In the well-constrained inorganic precipitation experiment (De-Carlo et al., 2015; Holcomb et al., 2016), the incorporation of impurities is apparently directly affected by the saturation state (since precipitation rate depends directly on omega and partitioning of elements depends directly on precipitation rate). Now, partitioning of elements (e.g. Mg, Sr) does not only rely on precipitation rate, but on a suite of other parameters, including temperature, seawater composition, photosynthetic activity and salinity. Not to mention species-specific differences in chemical composition of the aragonite (or calcite). This implies that changes in the $\upsilon 1$ FWHM may change with Mg/Ca or Sr/Ca, but those are not always and only related to changes in seawater $\Omega$ in biogenic material. This does not defy the outcome of this study, but in my opinion does warrant a more careful discussion.

For example, the 'apparent' control of [CO32-] on $\upsilon 1$ FWHM as shown in figure 3 may be just as real as that of $\Omega$Ar. Despite the outcome of the statistical modelling, both these two parameters seem equal explanations for the observed change in Raman shift, because both these parameters are (in a very similar way) responsible for the concentration of impurities in the aragonite. Therefore, the last sentence of the caption of figure 3 is misleading and needs to be changed.

Using a similar reasoning, the coral's results do not necessarily reflect only (or even primarily) the saturation state of the fluid from which they calcify. The results are highly interesting, but the results from the inorganic precipitation experiments do not justify the interpretation spelled out by the authors. As an example: on page 10, lines 1-2 for example, the change in $\upsilon 1$ FWHM is claimed to solely reflect omega, which should be nuanced.

page 11, lines 6-8 When looking at the 6 experiments with varying temperature, only two experiments with different temperatures were conducted at the same $\Omega$ar, according to figure 2. Is this enough to conclude that $\upsilon 1$ FWHM is not sensitive to precipitation rate? Figure 2b still shows a certain decreasing trend with increasing precipitation rate, especially when taking the average of the different experiments. Again, future work

should be focused on a more decoupled experimental set-up, to also investigate the observed correlation with solid Mg/Ca and fluid [CO2− 3 ].

Page 12, lines 6-22: this may be a crucial paragraph since the authors describe here why the Mg/Ca of the calcifying fluid is not related to the $v1$ FWHM, but rather $\Omega$ controls the incorporation of impurities and hence the average C-O bond length. However, this does not mean that Mg/Ca of the calcifying fluid could not have an equal effect on the Raman spectrum and hence, the measured $v1$ FWHM (e.g. in corals) may reflect either (or a combination) of fluid Mg/Ca and $\Omega$.

Sections 4.3 and 4.4 may need to reflect another interpretation and rephrasing to avoid the suggestion that $v1$ FWHM directly and only reflects $\Omega$.

In addition, there are some minor issues that I listed below that may help to further improve the manuscript.

page 2, line 17: saturation state is not the only thing that determines growth rates. Presence of inhibitors (Reddy et al., 2012. J Cryst Growth 352: 151-154) and the Ca:CO32- stoichiometry (Nehrke et al., 2007. Geochim Cosmochim Acta 71: 2240-2249) also affect growth rates.

page 2, lines 17-22: this is a bit of a stretch. First, the authors acknowledge that it is not certain ("if one exists") to what extent the internal and external saturation states are related. Therefore, knowing the internal saturation state does not necessarily result in an accurate forecast of the fate of coral calcification (even when ignoring the response of coral biomineralization to other environmental changes). Please rephrase.

―――――――――――――――

---

## Author Comment (AC2) · 26 Sep 2017

I have carefully read the manuscript by DeCarlo and others that describes the use of Raman spectroscopy to determine conditions at which (biogenic) calcium carbonate is precipitated. Overall, this manuscript contains an impressive amount of data, including results from inorganic experiments, culture experiment and field data. I have one serious concern about the (in)directness of the relation between $\nu_1$ FWHM and the saturation state, that needs revising before this manuscript can be accepted for publication in Biogeosciences.

**We thank the reviewer for carefully assessing our manuscript. The comments helped us to clarify how we conclude that $\Omega_{Ar}$ in the primary driver of $\nu_1$ FWHM**

in aragonite. **We added discussion of a subset of the abiogenic experiments with fluid [Ca$^{2+}$] manipulation, which decoupled $\Omega_{Ar}$ from Mg/Ca and [CO$_3^{2-}$]. We are confident that our revisions satisfy the issues raised by the reviewer, and doing so has improved our manuscript.**

*Comment 1:* The correlation between the Raman shift and the $\Omega_{Ar}$ may well be specific to the (inorganic) experiment and may not be general. As the authors explain in the introduction and methods, the actual shift in the spectrum is caused by other elements (impurities) or lattice distortions. In the well-constrained inorganic precipitation experiment (DeCarlo et al., 2015; Holcomb et al., 2016), the incorporation of impurities is apparently directly affected by the saturation state (since precipitation rate depends directly on omega and partitioning of elements depends directly on precipitation rate). Now, partitioning of elements (e.g. Mg, Sr) does not only rely on precipitation rate, but on a suite of other parameters, including temperature, seawater composition, photosynthetic activity and salinity. Not to mention species-specific differences in chemical composition of the aragonite (or calcite). This implies that changes in the $\nu_1$ FWHM may change with Mg/Ca or Sr/Ca, but those are not always and only related to changes in seawater $\Omega_{Ar}$ in biogenic material. This does not defy the outcome of this study, but in my opinion does warrant a more careful discussion.

*Response 1:* **The reviewer makes a good point that if trace element impurities (Mg/Ca and Sr/Ca) control the Raman $\nu_1$ FWHM, then there could be differences between abiogenic aragonites and corals. However, our abiogenic results do not point to substantial effects of either Mg/Ca or Sr/Ca on $\nu_1$ FWHM. In addition to our regression models, key data for decoupling Mg/Ca and $\Omega_{Ar}$ controls are two experiments conducted with elevated [Ca$^{2+}$] concentrations (f08 and g13), which reduced Mg/Ca but increased $\Omega_{Ar}$. The $\nu_1$ FWHM of these aragonites fall off the trend with Mg/Ca (revised Figure 3a and new Figure 5b), but are consistent with the $\Omega_{Ar}$ calibration (revised Figure 2 and new Figure 5a). We did not make the importance of these two experiments clear in the original manuscript, but we now**

highlight these results on page 12 lines 11-21. The Sr/Ca ratios of our abiogenic aragonites are primarily controlled by temperature (see Figure 3 in DeCarlo et al., 2015). If Sr/Ca influenced $\nu_1$ FWHM, we should see a temperature effect with experiments conducted at 20 °C or 40 °C falling away from our calibration $\Omega_{Ar}$-FWHM calibration, but this is not the case (Figure 2; page 12 lines 2-3), nor is there any correlation between Sr/Ca and $\nu_1$ FWHM (Fig. R2-1 below). It is also important to recognize that the ranges of Mg/Ca and Sr/Ca (as well as other trace elements) in our abiogenic aragonites are comparable or larger than those found in most tropical scleractinians.

The absence of clear effects of element/Ca ratios on $\nu_1$ FWHM suggests that increases in $\nu_1$ FWHM with $\Omega_{Ar}$ are probably driven primarily by disorder of $CO_3$ in the lattice. Although we do not yet have data to show exactly how this occurs on the molecular level, there may be a shift from highly crystalline aragonites forming at low $\Omega_{Ar}$ to relatively disordered or more amorphous-like aragonites forming at higher $\Omega_{Ar}$. We included a thorough discussion (section 4.1) of the potential effects of Mg/Ca (and now Sr/Ca) because changes in FWHM of calcite and amorphous calcium carbonate are often attributed to Mg/Ca. Yet there is no evidence that this is the case for aragonite. We agree with the reviewer that translating any relationships from abiogenic experiments to corals must be done with caution, and we have acknowledged this in the text (page 13 lines 17-18, page 18 lines 1-6). However, our proposed FWHM-$\Omega_{Ar}$ calibration appears to be a general process of aragonite precipitating from seawater, independent of minor changes in trace element composition. We have no reason to think this is any less generalizable than using abiogenic element partitioning coefficients to interpret biogenic carbonates, and our testing with JCp-1 demonstrates the close agreement between Raman and trace element geochemistry. Nevertheless, we added to the Conclusion section that additional tests on corals will be useful. For example, checking for correlations between $\nu_1$ FWHM and wavenumber is one way to test if element/Ca variability influences $\nu_1$ FWHM of coral skeletons

**(page 18 lines 2-6).**

*Comment 2:* For example, the 'apparent' control of $[CO_3^{2-}]$ on FWHM as shown in figure 3 may be just as real as that of $\Omega_{Ar}$. Despite the outcome of the statistical modelling, both these two parameters seem equal explanations for the observed change in Raman shift, because both these parameters are (in a very similar way) responsible for the concentration of impurities in the aragonite. Therefore, the last sentence of the caption of figure 3 is misleading and needs to be changed.

*Response 2:* **The reviewer is correct that variability in $[CO_3^{2-}]$ is difficult to isolate from $\Omega_{Ar}$. While these two parameters were strongly correlated when comparing across all the experiments in our abiogenic study, the two experiments conducted at elevated $[Ca^{2+}]$ provide some independence (page 12 lines 11-21 and new Figure 5c). Like the discussion of Mg/Ca above, FWHM of these two experiments fall off the $[CO_3^{2-}]$ trend, but are consistent with our FWHM-$\Omega_{Ar}$ calibration (new Figure 5). Thus, while our results indicate that $\Omega$ is truly the controlling factor, we agree with the reviewer that $[CO_3^{2-}]$ should not be dismissed entirely, and we have revised our discussion to leave room for other possibilities (page 12 lines 18-21, page 13 lines 17-18, page 18 lines 1-6, and the Figure 3 caption is revised to "... patterns observed here may be artefacts...").**

*Comment 3:* Using a similar reasoning, the coral's results do not necessarily reflect only (or even primarily) the saturation state of the fluid from which they calcify. The results are highly interesting, but the results from the inorganic precipitation experiments do not justify the interpretation spelled out by the authors. As an example: on page 10, lines 1-2 for example, the change in $\nu_1$ FWHM is claimed to solely reflect omega, which should be nuanced.

*Response 3:* **We agree that some of the language in the original manuscript was overly strong, but nevertheless we stand by our interpretation that coral FWHM is likely controlled by $\Omega_{Ar}$ for the reasons listed above in responses 1-2.**

**The reviewer is correct that other factors could theoretically have some effects (acknowledged on page 12 lines 18-21, page 13 lines 17-18, page 18 lines 1-6), but we have no evidence to support any factor other than $\Omega_{Ar}$ controlling FWHM of abiogenic aragonites or corals. We revised the manuscript to more clearly explain our reasoning that $\Omega_{Ar}$ is the controlling factor (section 4.1), while also stating that additional tests will be helpful to identify if there are subtle effects of other variables (added to page 12 lines 18-21, page 18 lines 2-6). Furthermore, we revised the description of the coral results (including the lines noted by the reviewer) to first state the changes in FWHM that were observed, and then we provide the changes in $\Omega_{Ar}$ assuming the abiogenic calibration applies to corals (page 10).**

*Comment 4:* page 11, lines 6-8 When looking at the 6 experiments with varying temperature, only two experiments with different temperatures were conducted at the same $\Omega_{Ar}$, according to figure 2. Is this enough to conclude that $\nu_1$ FWHM is not sensitive to precipitation rate? Figure 2b still shows a certain decreasing trend with increasing precipitation rate, especially when taking the average of the different experiments. Again, future work should be focused on a more decoupled experimental set-up, to also investigate the observed correlation with solid Mg/Ca and fluid [$CO_3^{2-}$].

*Response 4:* **The reviewer is correct that if taking the average of FWHM at each temperature, there would appear to be an inverse relationship to precipitation rate. However, since FWHM clearly increases with $\Omega_{Ar}$ (and precipitation rate) at a constant temperature, it would be very strange for precipitation rate to have both positive and negative effects on FWHM. That the data from all experiments fall along a single $\Omega_{Ar}$-FWHM calibration (Figure 2) strongly supports that $\Omega_{Ar}$ is the primary driver, not precipitation rate (Figure 3b).**

*Comment 5:* Page 12, lines 6-22: this may be a crucial paragraph since the authors describe here why the Mg/Ca of the calcifying fluid is not related to the $\nu_1$ FWHM, but rather controls the incorporation of impurities and hence the average C-O bond length.

[Figure]

However, this does not mean that Mg/Ca of the calcifying fluid could not have an equal effect on the Raman spectrum and hence, the measured $\nu_1$ FWHM (e.g. in corals) may reflect either (or a combination) of fluid Mg/Ca and $\Omega_{Ar}$.

*Response 5:* **We agree with the reviewer that this discussion is pivotal to interpreting the coral data. Our new discussion of experiments conducted at elevated [Ca$^{2+}$] adds to this argument. By changing the fluid Mg/Ca ratio, we achieved a factor ~2 change in aragonite Mg/Ca from ~5 to 2.5 mmol/mol but over a $\Omega_{Ar}$ range of only 10-12. This change in Mg/Ca covers most of the range of that found in shallow-water scleractinians (e.g., Gaetani et al., 2011 in GCA). However, there was no effect on FWHM. Thus, we can say that $\Omega_{Ar}$ was the primary driver of FWHM in the abiogenic aragonites (section 4.1 and response 1 above). Yet it is theoretically possible that Mg/Ca could cause subtle changes of FWHM in corals (page 13 lines 17-18, page 18 lines 1-6). This is a testable hypothesis because Mg/Ca will change FWHM and wavenumber in tandem (Bischoff et al., 1985; Perrin et al., 2016), whereas changes in FWHM driven by $\Omega_{Ar}$ are not associated with changes in $\nu_1$ wavenumber. Thus, future studies on corals could check for positive correlations between $\nu_1$ FWHM and wavenumber as a means of controlling for potential Mg/Ca effects (page 18 lines 2-6).**

*Comment 6:* Sections 4.3 and 4.4 may need to reflect another interpretation and rephrasing to avoid the suggestion that $\nu_1$ FWHM directly and only reflects $\Omega_{Ar}$.

*Response 6:* **We revised the Results section to make it clear that FWHM was measured, and that we are interpreting these data in terms of the abiogenic $\Omega_{Ar}$-FWHM calibration.**

In addition, there are some minor issues that I listed below that may help to further improve the manuscript.

*Comment 7:* page 2, line 17: saturation state is not the only thing that determines growth rates. Presence of inhibitors (Reddy et al., 2012. J Cryst Growth 352: 151-154)

and the Ca:CO32- stoichiometry (Nehrke et al., 2007. Geochim Cosmochim Acta 71: 2240-2249) also affect growth rates.

*Response 7:* **We revised this statement to say that elevating $\Omega_{Ar}$ contributes to (rather than "drives") aragonite nucleation and growth (page 2 line 18).**

*Comment 8:* page 2, lines 17-22: this is a bit of a stretch. First, the authors acknowledge that it is not certain ("if one exists") to what extent the internal and external saturation states are related. Therefore, knowing the internal saturation state does not necessarily result in an accurate forecast of the fate of coral calcification (even when ignoring the response of coral biomineralization to other environmental changes). Please rephrase.

*Response 8:* **We removed this sentence entirely as it was not necessary for our message and we agree that the sensitivity of calcification is likely more complicated.**

**Caption to Fig. R2-1. Raman true $\nu_1$ FWHM plotted as a function of measured aragonite Sr/Ca. There is no significant correlation (p = 0.23, $r^2$ = 0.06).**

[Figure]

Fig. 1.

[Figure]

[Figure]

---

## Author Response (AR1)

**Dear Editor,**

**Thank you for providing us the opportunity to revise our manuscript, "Coral calcifying fluid aragonite saturation states derived from Raman spectroscopy". Comments from the two reviewers highlighted several areas of the manuscript where additional details and explanations were required. We are confident that in revising according to the reviewers' suggestions that we improved the quality of our manuscript. Below are responses to each comment, including specific references to the changes we made in the manuscript.**

Reviewer 1:

*Comment 1:* The submitted manuscript of DeCarlo et al. reports interesting innovative research results regarding the use of Raman spectroscopy for the determination of aragonite saturation state in inorganic experiments and during coral calcification. As such it is timely, of broad scientific interest and fits into the scope of BG. I would expect this material to be publishable after careful revision.

Before explaining some of my concerns I need to underline I am not at all a Raman spectroscopy expert. Thus, Raman-specific technical details need to be reviewed by a respective expert before any decision on publication can be made.

In general I like this manuscript for it's interesting approach which warrents publication in my opinion. Nevertheless, some conclusions, I think, should be presented more carefully, underlining the potential and open questions of this approach.

*Response 1:* **We thank the reviewer for the supportive comments and careful consideration of our manuscript. We appreciate the issues raised, which highlight a couple areas of the manuscript that require clarification. Below, we respond to the reviewer's specific comments. While our results and conclusions remain the same, we agree with the reviewer that we needed to add a few details to further explain how the calculations were performed.**

*Comment 2:* It appears strange to me, the calibration (inorganic) yields results for the coral which are presented as correct reconstruction of internal cf saturation state. The calibration in my opinion is not entirely correct or it is not very clearly explained. I'd tried to recalculate the regression based on the data provided in the supplements. It appears, the uncertainties of the saturation state data from the experiment have not been included in the uncertainty estimate of the calibration. It rather looks like the means of FWHM plotted vs. Omega and a log-fit applied. This is critical as later the FWHM is used to reconstruct Omega. I hope, the point is clear, it would need quite the opposite plot and fit, Omega vs. FWHM for a calibartion useful for the desired calculation. Well, the fit obviousely, changes in this case.

*Response 2:* **We thank the reviewer for this comment, which alerted us to a point that we needed clarification. The reviewer is correct that our calibration is based on fitting FWHM to the log of $\Omega_{Ar}$ (see Table 2). We use $\Omega_{Ar}$ as the independent variable (x-axis in Figure 2) and FWHM as the dependent variable (y-axis in Figure 2) because our proposed mechanism is that $\Omega_{Ar}$ causes changes in FWHM (see section 4.1). However, we agree with the reviewer that this type of regression does not take errors in $\Omega_{Ar}$ into account, and that an alternative approach would be to fit $\Omega_{Ar}$ as a function of FWHM. We tried this method using the measured data from the WITec instrument (Figure R1-1 below, new Figure S6, and page 7 lines 27-28). The difference between the two approaches is very small, *e.g.* approximately 0.1 $\Omega_{Ar}$ unit difference for JCp-1.**

*Comment 3:* In any way, I could not replicate the Jcp-1 Omega of 12.3 with either of the calibrations. Could it be, each individual Raman result had been converted into a result for Omega and those results have been used to calculate an average of 12.3? If so, I did not get this from the manuscript... The Omega calculated from the mean FWHM would be >13, if I did the calculations right.

*Response 3:* **While we cannot reproduce the $\Omega_{Ar}$ of 13 stated by the reviewer, we agree that these calculations were not described clearly enough in the original manuscript. The reviewer is correct that we converted FWHM to $\Omega_{Ar}$ for each of the 440 JCp-1 measurements, and then we took the average of these $\Omega_{Ar}$ values. This is clarified in the revised manuscript (page 8 lines 13-14 and page 10 lines 1-2). Additionally, we clarified that the JCp-1 analyses were conducted on the WITec instrument (page 8 line 13), as this may have been the source of confusion.**

**We also wish to point out that we have posted the .R scripts to repeat all the analyses in the manuscript and reproduce the figures at the following link:**

**https://codeocean.com/2017/06/30/code-for-quot-coral-calcifying-fluid-aragonite-saturation-states-derived-from-raman-spectroscopy-quot/code**

**The calculations that produce JCp-1 $\Omega_{Ar}$ of 12.3 ± 0.3 from the raw data are there.**

*Comment 4:* Considerring the large uncertainties of the source data (Omega from inorganic experiments), and the fact, that corals do not perform such experiments to grow their skeleton, it appears not realistc to claim the precise reconstruction of coral cf Omega +-1-2, as stated in the abstract. Despite this critique I am confident and look forward to see this published as a paper in BG.

*Response 4:* **We agree with the reviewer that there are substantial uncertainties in $\Omega_{Ar}$ over the course of each abiogenic experiment, and this is discussed in detail on page 8. Nevertheless, the statistics of the calibration support our claim that $\Omega_{Ar}$ can be derived within 1-2 units with proper instrument configurations (page 1 lines 10-11 and Supplement). Furthermore, we have added more discussion of how we identified $\Omega_{Ar}$ as the primary driver of FWHM (section 4.1). We also acknowledge some uncertainty in how this can be applied to corals (page 13 lines 17-19, page 18 lines 1-6), although the available evidence does support the accuracy of our approach for deriving calcifying fluid $\Omega_{Ar}$.**

[Figure]

**Figure R1-1 (also new Figure S6). Measured Raman FWHM plotted against fluid $\Omega_{Ar}$ for the abiogenic aragonites analysed with the WITec instrument. The red curve shows the calibration when FWHM is fit as a function of $\Omega_{Ar}$, and the blue curve shows the calibration when $\Omega_{Ar}$ is fit as a function of FWHM.**

Reviewer 2:

I have carefully read the manuscript by DeCarlo and others that describes the use of Raman spectroscopy to determine conditions at which (biogenic) calcium carbonate is precipitated. Overall, this manuscript contains an impressive amount of data, including results from inorganic experiments, culture experiment and field data. I have one serious concern about the (in)directness of the relation between $v_1$ FWHM and the saturation state, that needs revising before this manuscript can be accepted for publication in Biogeosciences.

**We thank the reviewer for carefully assessing our manuscript. The comments helped us to clarify how we conclude that Ω in the primary driver of $v_1$ FWHM in aragonite. We added discussion of a subset of the abiogenic experiments with fluid $[Ca^{2+}]$ manipulation, which decoupled Ω from Mg/Ca and $[CO_3^{2-}]$. We are confident that our revisions satisfy the issues raised by the reviewer, and doing so has improved our manuscript.**

*Comment #1:* The correlation between the Raman shift and the Ω may well be specific to the (inorganic) experiment and may not be general. As the authors explain in the introduction and methods, the actual shift in the spectrum is caused by other elements (impurities) or lattice distortions. In the well-constrained inorganic precipitation experiment (DeCarlo et al., 2015; Holcomb et al., 2016), the incorporation of impurities is apparently directly affected by the saturation state (since precipitation rate depends directly on omega and partitioning of elements depends directly on precipitation rate). Now, partitioning of elements (e.g. Mg, Sr) does not only rely on precipitation rate, but on a suite of other parameters, including temperature, seawater composition, photosynthetic activity and salinity. Not to mention species-specific differences in chemical composition of the aragonite (or calcite). This implies that changes in the $v_1$ FWHM may change with Mg/Ca or Sr/Ca, but those are not always and only related to changes in seawater Ω in biogenic material. This does not defy the outcome of this study, but in my opinion does warrant a more careful discussion.

*Response #1:* **The reviewer makes a good point that if trace element impurities (Mg/Ca and Sr/Ca) control the Raman $v_1$ FWHM, then there could be differences between abiogenic aragonites and corals. However, our abiogenic results do not point to substantial effects of either Mg/Ca or Sr/Ca on $v_1$ FWHM. In addition to our regression models, key data for decoupling Mg/Ca and Ω controls are two experiments conducted with elevated $[Ca^{2+}]$ concentrations ("f08" and "g13"), which reduced Mg/Ca but increased Ω. The $v_1$ FWHM of these aragonites fall off the trend with Mg/Ca (revised Figure 3a and new Figure 5b), but are consistent with the Ω calibration (revised Figure 2 and new Figure 5a). We did not make the importance of these two experiments clear in the original manuscript, but we now highlight these results on page 12 lines 11-21. The Sr/Ca ratios of our abiogenic aragonites are primarily controlled by temperature (see Figure 3 in DeCarlo et al., 2015). If Sr/Ca influenced $v_1$ FWHM, we should see a temperature effect with experiments conducted at 20 °C or 40 °C falling away from our calibration Ω-FWHM calibration, but this is not the case (Figure 2; page 12 lines 2-3), nor is there any correlation between Sr/Ca and $v_1$ FWHM (Fig. R2-1 below). It is also important to recognize that the ranges of Mg/Ca and Sr/Ca (as well as other trace elements) in our abiogenic aragonites are comparable or larger than those found in most tropical scleractinians.**

The absence of clear effects of element/Ca ratios on $v_1$ FWHM suggests that increases in $v_1$ FWHM with $\Omega$ are probably driven primarily by disorder of $CO_3$ in the lattice. Although we do not yet have data to show exactly how this occurs on the molecular level, there may be a shift from highly crystalline aragonites forming at low $\Omega$ to relatively disordered or more amorphous-like aragonites forming at higher $\Omega$. We included a thorough discussion (section 4.1) of the potential effects of Mg/Ca (and now Sr/Ca) because changes in FWHM of calcite and amorphous calcium carbonate are often attributed to Mg/Ca. Yet there is no evidence that this is the case for aragonite. We agree with the reviewer that translating any relationships from abiogenic experiments to corals must be done with caution, and we have acknowledged this in the text (page 13 lines 17-18, page 18 lines 1-6). However, our proposed FWHM-$\Omega$ calibration appears to be a general process of aragonite precipitating from seawater, independent of minor changes in trace element composition. We have no reason to think this is any less generalizable than using abiogenic element partitioning coefficients to interpret biogenic carbonates, and our testing with JCp-1 demonstrates the close agreement between Raman and trace element geochemistry. Nevertheless, we added to the Conclusion section that additional tests on corals will be useful. For example, checking for correlations between $v_1$ FWHM and wavenumber is one way to test if element/Ca variability influences $v_1$ FWHM of coral skeletons (page 18 lines 2-6).

[Figure]

**Fig. R2-1. Raman true $v_1$ FWHM plotted as a function of measured aragonite Sr/Ca. There is no significant correlation (p = 0.23, $r^2$ = 0.06).**

*Comment #2:* For example, the 'apparent' control of $[CO_3^{2-}]$ on $v_1$ FWHM as shown in figure 3 may be just as real as that of $\Omega$. Despite the outcome of the statistical modelling, both these two parameters seem equal explanations for the observed change in Raman shift, because both these parameters are (in a very similar way) responsible for the concentration of impurities in the aragonite. Therefore, the last sentence of the caption of figure 3 is misleading and needs to be changed.

*Response #2:* **The reviewer is correct that variability in $[CO_3^{2-}]$ is difficult to isolate from $\Omega$. While these two parameters were strongly correlated when comparing across all the experiments in our abiogenic study, the two experiments conducted at elevated $[Ca^{2+}]$ provide some independence (page 12 lines 11-21 and new Figure 5c). Like the discussion of Mg/Ca above, FWHM of these two experiments fall off the $[CO_3^{2-}]$ trend, but are consistent with our FWHM-$\Omega$ calibration (new Figure 5). Thus, while our results indicate that $\Omega$ is truly the controlling factor, we agree with the reviewer that $[CO_3^{2-}]$ should not be dismissed entirely, and we have revised our discussion to leave room for other possibilities (page 12 lines 18-21, page 13 lines 17-18, page 18 lines 1-6, and the Figure 3 caption is revised to "… patterns observed here *may be* artefacts…").**

*Comment #3:* Using a similar reasoning, the coral's results do not necessarily reflect only (or even primarily) the saturation state of the fluid from which they calcify. The results are highly interesting, but the results from the inorganic precipitation experiments do not justify the interpretation spelled out by the authors. As an example: on page 10, lines 1-2 for example, the change in $v_1$ FWHM is claimed to solely reflect omega, which should be nuanced.

*Response #3:* **We agree that some of the language in the original manuscript was overly strong, but nevertheless we stand by our interpretation that coral FWHM is likely controlled by $\Omega$ for the reasons listed above in responses #1-2. The reviewer is correct that other factors could theoretically have some effects (acknowledged on page 12 lines 18-21, page 13 lines 17-18, page 18 lines 1-6), but we have no evidence to support any factor other than $\Omega$ controlling FWHM of abiogenic aragonites or corals. We revised the manuscript to more clearly explain our reasoning that $\Omega$ is the controlling factor (section 4.1), while also stating that additional tests will be helpful to identify if there are subtle effects of other variables (added to page 12 lines 18-21, page 18 lines 2-6). Furthermore, we revised the description of the coral results (including the lines noted by the reviewer) to first state the changes in FWHM that were observed, and then we provide the changes in $\Omega$ assuming the abiogenic calibration applies to corals (page 10).**

*Comment #4:* page 11, lines 6-8 When looking at the 6 experiments with varying temperature, only two experiments with different temperatures were conducted at the same $\Omega$, according to figure 2. Is this enough to conclude that $v_1$ FWHM is not sensitive to precipitation rate? Figure 2b still shows a certain decreasing trend with increasing precipitation rate, especially when taking the average of the different experiments. Again, future work should be focused on a more decoupled experimental set-up, to also investigate the observed correlation with solid Mg/Ca and fluid $[CO_3^{2-}]$.

*Response #4:* **The reviewer is correct that if taking the average of FWHM at each temperature, there would appear to be an inverse relationship to precipitation rate. However, since FWHM clearly increases with $\Omega$ (and precipitation rate) at a constant temperature, it would be very strange for precipitation rate to have both positive and negative effects on FWHM. That the data from all experiments fall along a single $\Omega$-FWHM calibration (Figure 2) strongly supports that $\Omega$ is the primary driver, not precipitation rate (Figure 3b).**

*Comment #5:* Page 12, lines 6-22: this may be a crucial paragraph since the authors describe here why the Mg/Ca of the calcifying fluid is not related to the $\nu_1$ FWHM, but rather controls the incorporation of impurities and hence the average C-O bond length. However, this does not mean that Mg/Ca of the calcifying fluid could not have an equal effect on the Raman spectrum and hence, the measured $\nu_1$ FWHM (e.g. in corals) may reflect either (or a combination) of fluid Mg/Ca and $\Omega$.

**Response #5: We agree with the reviewer that this discussion is pivotal to interpreting the coral data. Our new discussion of experiments conducted at elevated [Ca$^{2+}$] adds to this argument. By changing the fluid Mg/Ca ratio, we achieved a factor ~2 change in aragonite Mg/Ca from ~5 to 2.5 mmol/mol but over a $\Omega$ range of only 10-12. This change in Mg/Ca covers most of the range of that found in shallow-water scleractinians (*e.g.,* Gaetani et al., 2011 in *GCA*). However, there was no effect on FWHM. Thus, we can say that $\Omega$ was the primary driver of FWHM in the abiogenic aragonites (section 4.1 and response #1 above). Yet it is theoretically possible that Mg/Ca could cause subtle changes of FWHM in corals (page 13 lines 17-18, page 18 lines 1-6). This is a testable hypothesis because Mg/Ca will change FWHM and wavenumber in tandem (Bischoff et al., 1985; Perrin et al., 2016), whereas changes in FWHM driven by $\Omega$ are not associated with changes in $\nu_1$ wavenumber. Thus, future studies on corals could check for positive correlations between $\nu_1$ FWHM and wavenumber as a means of controlling for potential Mg/Ca effects (page 18 lines 2-6).**

*Comment #6:* Sections 4.3 and 4.4 may need to reflect another interpretation and rephrasing to avoid the suggestion that $\nu_1$ FWHM directly and only reflects $\Omega$.

**Response #6: We revised the Results section to make it clear that FWHM was measured, and that we are interpreting these data in terms of the abiogenic $\Omega$-FWHM calibration.**

In addition, there are some minor issues that I listed below that may help to further improve the manuscript.

*Comment #7:* page 2, line 17: saturation state is not the only thing that determines growth rates. Presence of inhibitors (Reddy et al., 2012. J Cryst Growth 352: 151-154) and the Ca:CO32-stoichiometry (Nehrke et al., 2007. Geochim Cosmochim Acta 71: 2240-2249) also affect growth rates.

**Response #7: We revised this statement to say that elevating $\Omega$ contributes to (rather than "drives") aragonite nucleation and growth (page 2 line 18).**

*Comment #8:* page 2, lines 17-22: this is a bit of a stretch. First, the authors acknowledge that it is not certain ("if one exists") to what extent the internal and external saturation states are related. Therefore, knowing the internal saturation state does not necessarily result in an accurate forecast of the fate of coral calcification (even when ignoring the response of coral biomineralization to other environmental changes). Please rephrase.

**Response #8: We removed this sentence entirely as it was not necessary for our message and we agree that the sensitivity of calcification is likely more complicated.**

---

## Author Response (AR2)

**Dear Dr. de Nooijer,**

**Thank you for providing constructive comments and offering us the opportunity to revise our manuscript. We have revised the manuscript following your suggestions, as detailed below. We greatly appreciate your time invested in editing our manuscript, and we look forward to publishing in Biogeosciences.**

**Sincerely,**
**Thomas DeCarlo on behalf of all co-authors**

Editor comments:

Dear Dr. DeCarlo and co-authors,

Thank you for uploading the revised version of your manuscript. In general, you have uploaded a considerably revised manuscript in which most issues raised by the reviewers have been addressed. Some minor issues remain that need to be corrected before this manuscript can be accepted for publication in Biogeosciences. In addition, I would like to ask you to have a critical look at your title: perhaps is does not (100%) reflect the content anymore.

Sincerely,

Lennart de Nooijer

**We followed the suggested minor revisions listed below. However, we prefer to retain the present title. We agree that our previous revisions highlight some additional investigations that will help to further understand the Raman- $\Omega_{Ar}$ proxy, but nevertheless our study does suggest that calcifying fluid $\Omega_{Ar}$ can be derived from Raman spectroscopy. Thus, we believe the present title reflects the main thrust of the paper.**

Page 18, line 16-18: "Finally, our analysis indicates that the Raman $\nu_1$ FWHM is an accurate proxy of fluid $\Omega_{Ar}$, making it a complementary approach to B/Ca and $\delta^{11}B$ because combining information from the two approaches allows calculating the full carbonate system via [$CO_3^{2-}$], pH and now importantly [$Ca^{2+}$].". The authors should be careful when making these kind of statements. The results from the inorganic precipitation experiments do not justify this interpretation promoted by the authors. In my opinion, it is currently a potential proxy for fluid $\Omega_{Ar}$ in inorganic aragonite, which can be used to develop a way to indirectly measure $\Omega_{Ar}$ of the calcification fluid of aragonite producing organisms, like corals.

**We revised this statement to, "Finally, our analysis of JCp-1 suggests that the Raman $\nu_1$ FWHM may be an accurate proxy of fluid $\Omega_{Ar}$, making it a complementary approach to B/Ca and $\delta^{11}B$ because combining information from the two approaches enables calculating the full carbonate system via [$CO_3^{2-}$], pH and possibly now [$Ca^{2+}$]$_{cf}$." We believe this better reflects the findings of the study, while leaving room for potential uncertainties that remain.**

Fig. 4. The authors seem to neglect the distinct offset between their observations and the ones made by Alkhatib and Eisenhauer (2017, GCA). How can you explain this offset?

**The difference is potentially related to the fluid media used in the experiments. We added two sentences, "Although our $K_D^{Mg/Ca}$ data show a similar trend to those of AlKhatib and Eisenhauer (2017), their $K_D^{Mg/Ca}$ are systematically lower. This offset is potentially explained by the different media used in the experiments: filtered seawater in DeCarlo et al. (2015) and Holcomb et al. (2016) as opposed to ammonium carbonate solutions in AlKhatib and Eisenhauer (2017)."**

Minor comments:
Please make change solid Mg/Ca to $Mg/Ca_{solid}$ or $Mg/Ca_{aragonite}$ to avoid confusion with media/seawater Mg/Ca.

**We changed to $Mg/Ca_{solid}$ throughout.**

Throughout the manuscript: $pCO_2$ instead of $pCO_2$. (i.e. please italicize the "p"). Also in the references please check $pCO_2$ and $CO_2$ (example page 25, lines 16 and 24).

**We changed to $pCO_2$ throughout, including the references.**